



# A closed-form model for layered snow slabs

Philipp Weißgraeber[1,*] and Philipp L. Rosendahl[2,*]

[1]University of Rostock, Chair of Lightweight Design, Germany
[2]Technical University of Darmstadt, Department of Civil and Environmental Engineering, Institute of Structural Mechanics and Design, Germany
[*]Contributed equally to this work.

**Correspondence:** Philipp L. Rosendahl (mail@2phi.de)

**Abstract.** We propose a closed-form analytical model for the mechanical behavior of stratified snow covers for the purpose of investigating and predicting the physical processes that lead to the formation of dry-snow slab avalanches. We represent the system of a stratified snow slab covering a collapsible weak layer by a beam composed of an arbitrary number of layers supported by an anisotropic elastic foundation in a two-dimensional plane-strain model. The model makes use of laminate
mechanics and provides slab deformations, stresses in the weak layer, and energy release rates of weak-layer anticracks in real time. The closed-form solution accounts for the layering-induced coupling of bending and extension in the slab and of shear and normal stresses in the weak layer. It is validated against experimentally recorded displacement fields and a comprehensive finite element model indicating very good agreement. We show that layered slabs cannot be homogenized into equivalent isotropic bodies and reveal the impact of layering on bridging with respect to weak-layer stresses and energy release rates. It is
demonstrated that inclined propagation saw tests allow for the determination of mixed-mode weak-layer fracture toughnesses. Our results suggest that such tests are dominated by mode I when cut upslope and comprise significant mode II contributions when cut downslope. A Python implementation of the presented model is publicly available as part of the *Weak Layer Anticrack Nucleation Model (WEAC)* software package under https://github.com/2phi/weac and https://pypi.org/project/weac (Rosendahl and Weißgraeber, 2022).

## 1 Introduction

Dry-snow slab avalanches are a critical danger in mountainous terrain with seasonal snow-covers. Not only because of temporal succession of meteorological events, such seasonal covers are composed of distinct individual layers. This yields snow covers that exhibit a stratification in terms of grain types, grain sizes, density, among others, and consequently also mechanical properties. Highly fragile layers (e.g., depth hoar or buried surface hoar) are referred to as weak layers and are known to
be the origin of slab avalanches (Bair, 2013). Their failure can lead to uncritical failure (*whumpf* sounds, shooting cracks) or avalanche release. The layering of snow covers is an essential part of avalanche forecasting (Richter et al., 2020) and for in-terrain decision making (Schweizer and Jamieson, 2007). It is known that the layering directly affects crack arrest or crack propagation (Birkeland et al., 2014). Hard layers within a snow slab have been identified as decisive for the effect of local load distribution within the snowpack (Schweizer et al., 1995; Camponovo and Schweizer, 1997).





Here, the so-called bridging effect that describes the load distribution through the slab onto lower layers as a function of slab and layer thicknesses, has been found an important feature of the mechanics of snow covers (Schweizer and Camponovo, 2001b; Schweizer and Jamieson, 2003). The effects appears differently in crack propagation, where thicker slabs are linked to larger avalanches, and onset of avalanche failure, where thinner slabs are more critical (Jamieson and Johnston, 1998; van Herwijnen and Jamieson, 2007). This is also discussed in the experimental and numerical study on stress fields below localized loadings by Thumlert and Jamieson (2014).

When snow cover models are linked to stability analyses (see Morin et al. (2020) for a comprehensive review), typically stability indices are used (McClung and Schweizer, 1999; Lehning et al., 2004). These indices typically employ strength-based methods such as the limit equilibrium method (Föhn, 1987; Huang, 2014). Often, stress fields are obtained by using solutions derived from the Boussinesq solution of an infinite half-plane under a point load (Föhn, 1987; Gaume and Reuter, 2017). Monti et al. (2015) proposed an equivalent-layer approach to allow for the use of solutions of isotropic continua for the stress analysis of layered slabs. Since the early works of Smith and Chu (1972) and Smith and Curtis (1975), finite element methods have been used to study stratified snowpacks (Schweizer, 1993; Habermann et al., 2008). These studies also clearly highlights the role of stratification and bridging on the stress and displacement fields within the snowpack.

The importance of bridging has been accounted for in the beam models by Heierli and Zaiser (2008) and Heierli (2008). Along with the concept of anticracks, these models allowed for an insight into avalanche release and gave a physical explanation for *whumpf* sounds and remote triggering of avalanches, both caused by the sudden expansion of a local weak-layer collapse. Based on these models we have proposed a refined beam model for the analysis of stresses and energy release rates of cracks in weak layers (Rosendahl and Weißgraeber, 2020a). However, the above models are restricted to homogeneous slabs. The role of bending on the collapse of weak layers was also studied by means of the discrete element method (Gaume et al., 2015; Bobillier et al., 2018). Gaume et al. (2018) studied weak-layer collapse by means of an elastoplastic material model accounting for softening and volume reduction. Studying the effect of the slab properties on crack initiation and propagation, van Herwijnen and Jamieson (2007), Sigrist and Schweizer (2007), Habermann et al. (2008), and Reuter et al. (2015) have addressed the role of layering on fracture within snow packs.

The importance of fracture mechanics for the analysis of avalanche release has been emphasized by many researchers (McClung, 1979, 1981; Heierli and Zaiser, 2006; Sigrist and Schweizer, 2007; Gauthier and Jamieson, 2008) and the significance of the fracture energy as the decisive material property has been highlighted (McClung and Schweizer, 2006; McClung, 2007; Heierli et al., 2008). In fracture mechanics models the energy balance of propagating cracks is considered as the central condition for the analysis of avalanche release. Using Föhn's solution (Föhn, 1987) and the empirical measure of a critical crack length (Gaume et al., 2017), Gaume and Reuter (2017) have proposed to link strength-based approaches and fracture mechanics approaches to assess the instability of snowpacks. Using an implicitly coupled stress and energy criterion we have proposed a failure model for anticrack initiation under mixed-mode loading that considers stresses and energy simultaneously (Rosendahl and Weißgraeber, 2020b).

In order to account for the crucial effect of layering on failure processes within a snowpack, we propose a new model for layered snow slabs on collapsible weak layers. In order to allow for efficient implementation in model chains and for use for



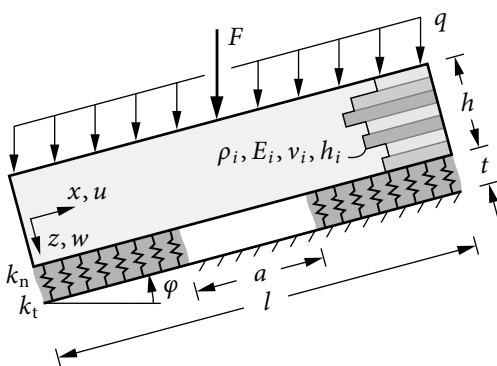

**Figure 1.** Stratified snowpack composed of an arbitrary number of slab layers and a weak layer modeled as an elastic foundation.

extensive parametric studies, a closed-form analytical solution is obtained by utilizing the concepts of mechanics of layered composites (Jones, 1998).

## 2 Mechanical model

In the present work, we model a stratified snow cover as a system comprised of i) a snow slab, represented by an arbitrarily layered beam, that rests ii) on a weak layer, represented by an elastic foundation. The beam kinematics and its constitutive

behavior are derived from first-order shear deformation theory of laminated plates under cylindrical bending (Reddy, 2003). The weak layer can be understood as an infinite set of smeared springs with normal and shear stiffness attached to the bottom side of the slab. This yields a system of fully coupled bending, extension and shear deformations of both slab and weak layer.

### 2.1 Governing equations

We consider a segment of the stratified snow pack on an inclined slope of angle $\varphi$ as shown in Fig. 1. As typical for beam

analyses, the axial coordinate $x$ points left-to-right along the beam midplane and is zero at its left end. The thickness coordinate $z$ is perpendicular to the midplane, points downwards and is zero at the center line. Slope angles $\varphi$ are counted positive about the $y$ axis of the right-handed Cartesian coordinate system (counterclockwise). Note that on inclined slopes ($\varphi \neq 0$), the axial and normal beam axes ($x$ and $z$) do not coincide with the horizontal and vertical directions.

The slab with total thickness $h$ is composed of $N$ layers with individual ply thicknesses $h_i = z_{i+1} - z_i$, each assumed

homogeneous and isotropic (Fig. 2). Young's modulus, Poisson's ratio and density of each layer are denoted by $E_i$, $\nu_i$ and $\rho_i$, respectively. The weak layer of thickness $t$ can be anisotropic and its normal and tangential stiffnesses are

$$k_{\mathrm{n}} = \frac{E'_{\mathrm{wl}}}{t}, \tag{1a}$$




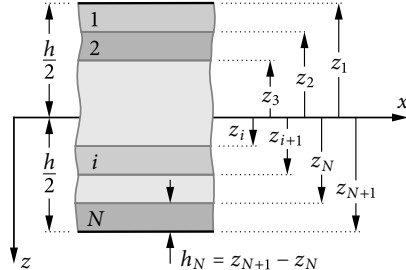

**Figure 2.** Slab of total thickness $h$ composed of $N$ individual layers. A layer $i$ is characterized by its height $h_i$ and its the top and bottom coordinates $z_i$ and $z_{i+1}$, respectively.

where $E'_{\mathrm{wl}} = E_{\mathrm{wl}}/(1 - \nu^2)$ is the weak layer's plane-strain elastic modulus and

$$k_{\mathrm{t}} = \frac{G_{\mathrm{wl}}}{t}, \tag{1b}$$

where $G_{\mathrm{wl}}$ is the weak layer's plane-strain shear modulus, respectively. The slab is loaded by its own weight, i.e., the gravitational load $q$, and an external load $F$ (e.g., a skier) in vertical direction. The gravity load corresponds to the sum of the weight of all layers

$$q = g \sum_{i=1}^{N} h_i \rho_i. \tag{2}$$

It is split into a normal component $q_{\mathrm{n}} = q \cos\varphi$ and a tangential component $q_{\mathrm{t}} = -q \sin\varphi$ that are introduced as line loads. The

tangential gravity line load acts at center of gravity in thickness direction

$$z_{\mathrm{s}} = \frac{\sum_{i=1}^{N} (z_i + z_{i+1}) h_i \rho_i}{2 \sum_{i=1}^{N} h_i \rho_i}, \tag{3}$$

in the slab, where $(z_i + z_{i+1})/2$ yields each layer's center $z$-coordinate. For relevant slab thicknesses the external load can be modeled as a point load and is introduced as a force with a normal component $F_{\mathrm{n}} = F \cos\varphi$ and a tangential component $F_{\mathrm{t}} = -F \sin\varphi$.

Deformations of the slab are described by means of the first-order shear deformation theory (FSDT) of laminated plates under cylindrical bending (Reddy, 2003). By dropping the Kirchhoff assumption of orthogonality of cross sections and midplane, this allows for the consideration of shear deformations. We consider midplane deflections $w_0$, midplane tangential displacements $u_0$ and the rotation $\psi$ of cross sections. The quantities define the displacement field of the beam according to

$$w(x,z) = w_0(x), \tag{4a}$$

$$u(x,z) = u_0(x) + z\psi(x). \tag{4b}$$

At the interface between slab and weak layer ($z = h/2$), the displacement fields of slab $(u,w)$ and weak-layer $(v,\omega)$ coincide. Using Eqs. (4a) and (4b), this yields $\bar{v} = \bar{u} = u_0 + \psi h/2$ and $\bar{\omega} = \bar{w} = w_0$, where the bar indicates quantities at the interface.



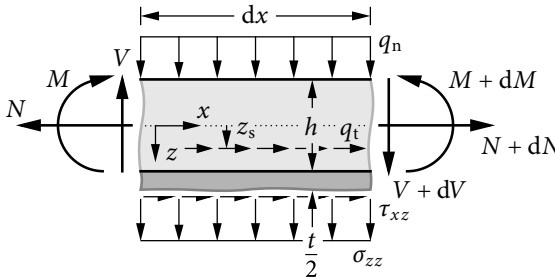

**Figure 3.** Free-body cut of an infinitesimal segment of length of the layered slab of height with half of the weak layer.

Modeling the weak layer as an elastic foundation of an infinite set of smeared linear elastic springs, yields constant strains and consequently a constant deformation gradient through its thickness. Hence, weak-layer stresses can be expressed through the differential deformation between the lower boundary of the weak layer ($v = \omega = 0$) and its deformations at the interface:

$$\sigma_{zz}(x) = E_{\mathrm{wl}}\varepsilon_{zz}(x) = E_{\mathrm{wl}}\frac{\mathrm{d}\omega(x,z)}{\mathrm{d}z} = E_{\mathrm{wl}}\frac{0 - \bar{\omega}(x)}{t}$$
$$= -k_{\mathrm{n}}w_0(x), \tag{5a}$$

$$\tau_{xz}(x) = G_{\mathrm{wl}}\gamma_{xz}(x) = G_{\mathrm{wl}}\left(\frac{\mathrm{d}v(x,z)}{\mathrm{d}z} + \frac{\mathrm{d}\omega(x,z)}{\mathrm{d}x}\right)$$
$$= G_{\mathrm{wl}}\left(\frac{0 - \bar{v}(x)}{t} + \frac{\bar{\omega}'(x)}{2}\right)$$
$$= k_{\mathrm{t}}\left(\frac{t}{2}w_0'(x) - u_0(x) - \frac{h}{2}\psi(x)\right). \tag{5b}$$

From the free body-cut of an infinitesimal beam section of the layered slab (Fig. 3), we obtain the equilibrium conditions of the section forces and moments:

$$0 = \frac{\mathrm{d}N(x)}{\mathrm{d}x} + \tau(x) + q_{\mathrm{t}}, \tag{6a}$$

$$0 = \frac{\mathrm{d}V(x)}{\mathrm{d}x} + \sigma(x) + q_{\mathrm{n}}, \tag{6b}$$

$$0 = \frac{\mathrm{d}M(x)}{\mathrm{d}x} - V(x) + \frac{h+t}{2}\tau(x) + z_{\mathrm{s}}q_{\mathrm{t}}. \tag{6c}$$

To connect the slab section forces (normal force $N$, shear force $V$, and bending moment $M$) to the deformations of the layered slab, we make use of the mechanics of composite laminates. First-order shear deformation theory of laminate plates under cylindrical bending yields

$$\begin{pmatrix} N(x) \\ M(x) \end{pmatrix} = \begin{pmatrix} A_{11} & B_{11} \\ B_{11} & D_{11} \end{pmatrix} \begin{pmatrix} u_0'(x) \\ \psi'(x) \end{pmatrix}, \tag{7a}$$

and

$$V(x) = \kappa A_{55}\left(w_0'(x) + \psi(x)\right). \tag{7b}$$





These constitutive equations contain the extensional stiffness $A_{11}$, the bending stiffness $D_{11}$, the bending–extension coupling stiffness $B_{11}$, and the shear stiffness $\kappa A_{55}$ of the layered slab. The coupling stiffness $B_{11}$ accounts for the bending–extension coupling of asymmetrically layered systems such as bimetal bars. These stiffness quantities are obtained by weighted[1] integration of the individual ply stiffness properties:

$$A_{11} = \int_{-h/2}^{h/2} \frac{E(z)}{1-\nu(z)^2}\,\mathrm{d}z = \sum_{i=1}^{N} \frac{E_i}{1-\nu_i^2}h_i, \tag{8a}$$

$$B_{11} = \int_{-h/2}^{h/2} \frac{E(z)}{1-\nu(z)^2}z\,\mathrm{d}z = \sum_{i=1}^{N} \frac{E_i}{1-\nu_i^2}z_i h_i, \tag{8b}$$

$$D_{11} = \int_{-h/2}^{h/2} \frac{E(z)}{1-\nu(z)^2}z^2\,\mathrm{d}z = \sum_{i=1}^{N} \frac{E_i}{1-\nu_i^2}\left(\frac{h_i^3}{12}+h_i z_i^2\right), \tag{8c}$$

$$A_{55} = \int_{-h/2}^{h/2} G(z)\,\mathrm{d}z = \sum_{i=1}^{N} G_i h_i. \tag{8d}$$

The shear correction factor $\kappa$ complements the shear stiffness $\kappa A_{55}$. It is set to $5/6$ as a good approximation for the layered slab of rectangular cross-section (Klarmann and Schweizerhof, 1993). The above quantities are given for the case of isotropic layers. Orthotropic layers can be considered following the same approach by using directional elastic properties of the individual layers instead of an isotropic Young's modulus.

In the special case of a homogeneous, isotropic slab with Young's modulus $E_{\mathrm{sl}}$ and Poisson's ratio $\nu$, the laminate stiffnesses take the homogeneous stiffness properties well-known from beam theory:

$$A_{11} = \frac{E_{\mathrm{sl}}h}{1-\nu^2}, \tag{9a}$$

$$D_{11} = \frac{E_{\mathrm{sl}}h^3}{12\left(1-\nu^2\right)}, \tag{9b}$$

$$A_{55} = \frac{E_{\mathrm{sl}}h}{2\left(1+\nu\right)}, \tag{9c}$$

and the coupling stiffness vanishes ($B_{11} = 0$).

## 2.2 System of differential equations and its solution

The equations of the kinematics of the weak layer, (5a) and (5b), the equilibrium conditions, (6a) to (6c), and the constitutive equations of the layered beam with first-order shear deformation theory, (7a) and (7b), provide a complete description of the mechanics of the layered snowpack and constitute a system of ordinary differential equations (ODEs) of second order.

---

[1]Weighted by the moment of area of the cross-section of zeroth, first, and second order.



Introducing the vector of unknown functions

$$\boldsymbol{z}(x) = \begin{bmatrix} u_0(x) & u_0'(x) & w_0(x) & w_0'(x) & \psi(x) & \psi'(x) \end{bmatrix}^\mathsf{T}, \tag{10}$$

the governing equations can be expressed as a first-order system of the form

$$\boldsymbol{z}'(x) = \boldsymbol{K}\boldsymbol{z}(x) + \boldsymbol{q}, \tag{11}$$

where bold upper-case symbols denote matrices and bold lower-case symbols indicate vectors. For the derivation of this ODE system and the definitions of the system matrix $\boldsymbol{K}$ and the right-hand side vector $\boldsymbol{q}$, see Appendix A.

The solution of the nonhomogeneous ODE system (11) is composed of a complementary solution vector $\boldsymbol{z}_\mathrm{h}(x)$ and a

particular integral vector $\boldsymbol{z}_\mathrm{p}$, where the latter is constant in the present case. The complementary solution can be obtained from an eigenanalysis of the system matrix $\boldsymbol{K}$. Depending on the layering and the material properties, $\boldsymbol{K}$ has six real or complex eigenvalues. Since the beam is bedded, it has no rigid body motions and all eigenvalues of nonzero. Real eigenvalues occur as sets of two eigenvalues with opposite signs $\pm\lambda_\mathbb{R}$ and linearly independent eigenvectors $\boldsymbol{v}_{\mathbb{R}\pm} \in \mathbb{R}^6$. Complex eigenvalues appear as complex conjugates $\lambda_\mathbb{C}^\pm = \lambda_\Re \pm i\lambda_\Im$ with the corresponding complex eigenvectors $\boldsymbol{v}_\mathbb{C}^\pm = \boldsymbol{v}_\Re \pm i\boldsymbol{v}_\Im$ such that $\boldsymbol{v}_\mathbb{C}^\pm \in \mathbb{C}^6$ and

$\boldsymbol{v}_\Re, \boldsymbol{v}_\Im \in \mathbb{R}^6$. Denoting the number of sets of real eigenvalue pairs as $N_\mathbb{R} \in \{0,\dots,3\}$ and the number of complex conjugate eigenvalue pairs as $N_\mathbb{C} \in \{0,\dots,3\}$ such that $N_\mathbb{R} + N_\mathbb{C} = 3$, the complementary solution is given by the linear combination

$$\begin{aligned}
\boldsymbol{z}_\mathrm{h}(x) = &\sum_{n=1}^{N_\mathbb{R}} C_{\mathbb{R}+}^{(n)} \exp\left(+\lambda_\mathbb{R}^{(n)} x\right) \boldsymbol{v}_{\mathbb{R}+}^{(n)} \\
&+ C_{\mathbb{R}-}^{(n)} \exp\left(-\lambda_\mathbb{R}^{(n)} x\right) \boldsymbol{v}_{\mathbb{R}-}^{(n)} \\
&+ \sum_{n=1}^{N_\mathbb{C}} C_\Re^{(n)} \exp\left(\lambda_\Re^{(n)} x\right) \left[\boldsymbol{v}_\Re^{(n)} \cos\left(\lambda_\Im^{(n)} x\right) \right. \\
&\qquad\qquad\qquad\qquad \left. - \boldsymbol{v}_\Im^{(n)} \sin\left(\lambda_\Im^{(n)} x\right)\right] \\
&+ C_\Im^{(n)} \exp\left(\lambda_\Re^{(n)} x\right) \left[\boldsymbol{v}_\Re^{(n)} \sin\left(\lambda_\Im^{(n)} x\right) \right. \\
&\qquad\qquad\qquad\qquad \left. + \boldsymbol{v}_\Im^{(n)} \cos\left(\lambda_\Im^{(n)} x\right)\right].
\end{aligned} \tag{12}$$

The particular solution is obtained using the method of undetermined coefficients, which yields the constant vector

$$\boldsymbol{z}_\mathrm{p} = \begin{bmatrix} \frac{q_\mathrm{t}}{k_\mathrm{t}} + \frac{h(h+t-2z_\mathrm{s})\,q_\mathrm{t}}{4\kappa A_{55}} & 0 & \frac{q_\mathrm{n}}{k_\mathrm{n}} & 0 & \frac{(2z_\mathrm{s}-h-t)\,q_\mathrm{t}}{2\kappa A_{55}} & 0 \end{bmatrix}^\mathsf{T}. \tag{13}$$

The general solution of the system

$$\boldsymbol{z}_\bullet(x) = \boldsymbol{z}_\mathrm{h}(x) + \boldsymbol{z}_\mathrm{p}, \tag{14}$$

comprises six unknown coefficients $C_\bullet^{(n)}$ that must be identified from boundary and transmission conditions. It can be given in the matrix form

$$\boldsymbol{z}_\bullet(x) = \boldsymbol{Z}_\mathrm{h}(x)\,\boldsymbol{c}_\bullet + \boldsymbol{z}_\mathrm{p}, \tag{15}$$

where $\boldsymbol{Z}_\mathrm{h} : \mathbb{R} \to \mathbb{R}^{6\times6}$ is a matrix-valued function with the summands of Eq. (12) as column vectors and $\boldsymbol{c}_\bullet \in \mathbb{R}^6$ a vector containing the six free constants $C_\bullet^{(n)}$ according of Eq. (12).





### 2.3 Layered segments without elastic foundation

If the slab is not supported by an elastic foundation (e.g., when the weak layer has collapsed or when a saw cut is introduced in
a propagation saw test), the general solution simplifies. In the equilibrium conditions (6a) to (6c), the normal and shear stress
terms are omitted since no stresses act on the bottom side of the slab. The constitutive equations (7a) and (7b) remain the same.
After some calculation (see Appendix B) one obtains the general solution of polynomials of fourth order. In matrix form, the
system reads

$$z_\circ(x) = \mathcal{P}(x)\,c_\circ + p(x),  \tag{16}$$

where $\mathcal{P}(x)$ and $p(x)$ are the polynomial matrix and vector, respectively. Again, a vector of six unknown coefficients

$$c_\circ = \begin{bmatrix} C_\circ^{(1)} & C_\circ^{(2)} & \dots & C_\circ^{(6)} \end{bmatrix}^\mathsf{T}.  \tag{17}$$

must be determined from boundary and transmission conditions.

### 2.4 Global system assembly

The general solutions presented above allow for the investigation of different systems composed of segments of supported and
unsupported layered slabs. Possible configurations of interest are, e.g., skier-loaded snowpacks, skier-loaded snowpacks with
a partially collapsed weak layer, or propagation saw test (PSTs) with an artificially introduced (sawed) edge crack. Assemblies
of such configurations are illustrated in Fig. 4.

Individual segments are connected through transmission conditions given in terms of displacements and section forces (see
Appendix C). Adding boundary conditions at the left and right ends of the beam, assembles the desired global system. Inserting
the general solutions (15) and (16) into the boundary and transmission conditions, yields equations that only depend on free
constants. The set of equations can be assembled into a system of linear equations with $k = 6N_\mathrm{b}$ degrees of freedom, where
$N_\mathrm{b}$ is the number of beam segments. In matrix form, the system reads

$$\Psi c = f.  \tag{18}$$

Here, $\Psi \in \mathbb{R}^{k \times k}$ is a square matrix of full rank, $c \in \mathbb{R}^k$ is the vector of all free constants, and $f \in \mathbb{R}^k$ is the right-hand-
side vector that contains the particular solutions and displacement discontinuities induced by concentrated loads. With only $k$
degrees of freedom, the system can be solved in real-time using standard methods such as Gaussian elimination or lower-upper
decomposition.

### 2.5 Computation of displacements, stresses and energy release rates

Substituting the coefficients $C^{(n)}$ obtained from Eq. (18) for each beam segment back into the general solutions (15) and (16),
yields the vector $z(x)$, which contains all slab displacement functions, see Eq. (10).

Inserting the slab deformation solution into Eqs. (5a) and (5b), provides weak-layer normal and shear stresses, respectively.
Note that weak interfaces do not allow for capturing highly localized stress concentrations (e.g., stress singularities) as they





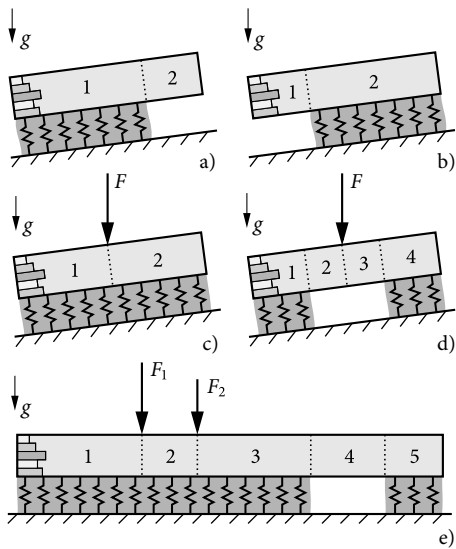

**Figure 4.** Exemplary systems of interest assembled from supported and unsupported layered slabs with numbered segments: a) downslope PST, b) upslope PST, c) skier-loaded snowpack, d) partially fractured weak-layer, and d) layered slab loaded by multiple skiers with partially fractured weak-layer. Dotted lines indicate transmission conditions for the continuity of displacements and section forces.

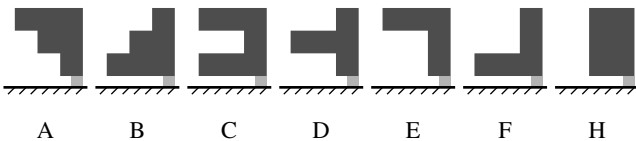

| A | B | C | D | E | F | H |

**Figure 5.** Illustration of benchmark snow profiles used in the present work. Material properties of hard, medium, and soft slab layers (dark) and the weak layer (light) are given in Table 1. The weak layer is $2\,\mathrm{cm}$ thick and the slab layers have a thickness of $12\,\mathrm{cm}$ each. Similar profiles were used by, e.g., Habermann et al. (2008) and Monti et al. (2015). Here, we complement the homogeneous slab H.

occur at crack tips. However, outside the immediate vicinity of crack tips, weak-interface kinematics provide accurate stress solution (Rosendahl and Weißgraeber, 2020a).

The total differential energy release rate of cracks in the weak layer $\mathcal{G}$ is composed of contributions from mode I (crack closure) and mode II (crack sling). Following Krenk (1992) it can be given as

$$\mathcal{G}(a) = \mathcal{G}_{\mathrm{I}}(a) + \mathcal{G}_{\mathrm{II}}(a) = \frac{\sigma(a)^2}{2k_{\mathrm{n}}} + \frac{\tau(a)^2}{2k_{\mathrm{t}}}, \tag{19}$$

where $a$ denotes the crack-tip coordinate. Energy release rates obtained using weak-interface kinematics cannot capture very short cracks but, again, provide accurate results for cracks of a certain minimum length (Hübsch et al., 2021).





**Table 1.** Considered snow layers and their elastic properties with reference to three-layer slabs used by Habermann et al. (2008).

| Layer | Hand hardness index | Density $\rho$ (kg/m$^3$) | Young's modulus $E$ (MPa) | Poisson's ratio $\nu$ |
|---|---|---|---|---|
| Hard | P | 350 | 93.8 | 0.25 |
| Medium | 1F | 270 | 30.0 | 0.25 |
| Soft | 4F | 180 | 5.0 | 0.25 |
| Weak layer | F– | 100 | 0.15 | 0.25 |

## 3 Model validation

With reference to the analysis of snowpack layering by Habermann et al. (2008) and Monti et al. (2015), we use three-layered slabs proposed as schematic hardness profiles by Schweizer and Wiesinger (2001), that are composed of soft, medium, and hard snow as benchmark slab configurations (Fig. 5). Assuming bonded slabs (e.g., rounded grains) and considering the density–hand hardness relations given by Geldsetzer and Jamieson (2000), we assume densities of $\rho = 350$, 270, and $180\,\mathrm{kg/m^3}$ for hard, medium, and soft snow layers with hand hardness indices pencil (P), four fingers (4F), and one finger (1F), respectively. From slab densities, we calculate the Young's modulus using the density-parametrization developed by Gerling et al. (2017) using acoustic wave propagation experiments and improved by Bergfeld et al. (2022) using full-field displacement measurements

$$E_{\mathrm{sl}}(\rho) = 6.5 \cdot 10^3\,\mathrm{MPa} \left( \frac{\rho}{\rho_0} \right)^{4.4}, \tag{20}$$

where $\rho_0 = 917\,\mathrm{kg/m^3}$ is the density of ice. Each slab layer is $12\,\mathrm{cm}$ thick and their individual material properties are given in Table 1. With reference to Jamieson and Schweizer (2000), who report weak layer thickness between $0.2$ and $3\,\mathrm{cm}$, we assume a weak-layer thickness of $t = 2\,\mathrm{cm}$. Following density measurements of surface hoar layers by Föhn (2001) who reports densities i) between $44$ and $215\,\mathrm{kg/m^3}$ with a mean of $102.5\,\mathrm{kg/m^3}$ and ii) between $75$ and $252\,\mathrm{kg/m^3}$ with a mean of $132.4\,\mathrm{kg/m^3}$ using two different measurement techniques, we assume a weak-layer density of $\rho_{\mathrm{wl}} = 100\,\mathrm{kg/m^3}$, and a Young's modulus of $E_{\mathrm{wl}} = 0.15\,\mathrm{MPa}$. Other parameters are summarized in Table 2.

### 3.1 Finite element reference model

To validate the model, in particular with respect to different slab layerings, we compare the analytical solution to finite element analyses (FEA). The finite element model is assembled from individual layers with unit out-of-plane width on an inclined slope. Each layer is discretized using at least 10 eight-node biquadratic plane-strain continuum elements with reduced integration through its thickness. The lowest layer corresponds to the weak layer and rests on a rigid foundation. Weak-layer cracks are introduced by removing all weak-layer elements on the crack length $a$. The mesh is refined towards stress concentration


**Table 2.** Material properties used throughout this work unless specified differently.

| Property | Symbol | Value |
|---|---|---|
| Skier weight | $m$ | $80\,\mathrm{kg}$ |
| Slope angle | $\varphi$ | $38\,^{\circ}$ |
| Slab thickness[*] | $h$ | $36\,\mathrm{cm}$ |
| Weak-layer thickness[*] | $t$ | $2\,\mathrm{cm}$ |
| Effective ouf-of-plane ski length | $l_\mathrm{o}$ | $100\,\mathrm{cm}$ |
| Young's modulus weak layer | $E_\mathrm{wl}$ | $0.15\,\mathrm{MPa}$ |
| Poisson's ratio | $\nu$ | $0.25$ |
| Length of PST block | $l_\mathrm{PST}$ | $250\,\mathrm{cm}$ |
| Length of PST cut | $a_\mathrm{PST}$ | $50\,\mathrm{cm}$ |

[*]Thicknesses given in slope-normal direction.

such as crack tips and convergence has been controlled carefully. The weight of the snowpack is introduced by providing the gravitational acceleration $g$ and assigning each layer its corresponding density $\rho$. The load introduced by a skier is modeled as a concentrated force acting on the top of the slab. If skier loading is considered, the horizontal dimensions of the model are

chosen large enough for all gradients to vanish. Typically $10\,\mathrm{m}$ suffice. Boundary conditions of PST experiments are free ends. In the FE model, the energy release rate of weak-layer cracks

$$\mathcal{G}_{\mathrm{FE}}(a) = -\frac{\partial\Pi(a)}{\partial a} \approx -\frac{\Pi(a+\Delta a) - \Pi(a-\Delta a)}{2\Delta a}, \tag{21}$$

is computed using the central difference quotient to approximate the first derivative of the total potential $\Pi$ with respect to $a$. The crack increment $\Delta a$ corresponds to the element size and could be increased twofold or threefold without impacting

computed values of $\mathcal{G}_{\mathrm{FE}}(a)$. Weak-layer stresses are evaluated in its vertical center.

### 3.2 Visualization of displacement and stress fields

Although visual representations of deformation and stress fields are limited to qualitative statements, they illustrate the principal responses of structures in different load cases. For this purpose, Fig. 6 compares principal stresses in a deformed slab-on-weak-layer system between present model and finite element reference solution. The system is loaded by the weight of the

homogeneous slab ▮ H and a concentrated force representing an $80\,\mathrm{kg}$ skier. Deformations are scaled by a factor of 200 and the weak-layer thickness by a factor of 4. In the slab, we show maximum principal normal stresses (tension) normalized to their tensile normal strength $\sigma_\mathrm{c}^+ = 9.1\,\mathrm{kPa}$ obtained from the scaling law

$$\sigma_\mathrm{c}^+(\rho) = 240\ \mathrm{kPa}\left(\frac{\rho}{\rho_0}\right)^{2.44}, \tag{22}$$



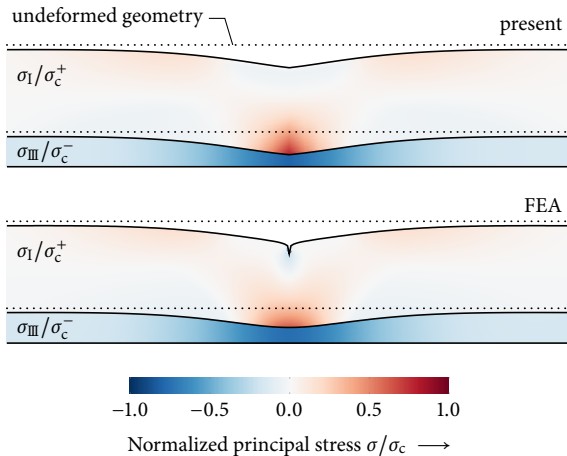

**Figure 6.** Principal stresses and 200 times scaled snowpack deformations in the central 200 cm section of a skier-loaded snowpack comparing the present model (top) and the FEA reference model (bottom). In the homogeneous slab ▮ H, maximum principal normal stresses $\sigma_{\mathrm{I}}$ (tension) normalized their tensile strength $\sigma_c^+ = 9.1\,\mathrm{kPa}$ are shown. In the weak layer we show minimum principal normal stresses $\sigma_{\mathrm{III}}$ (compression) normalized to an assumed weak layer compressive strength of $\sigma_c^- = 2.6\,\mathrm{kPa}$. The weak-layer thickness is scaled by a factor of 4 for illustration.

by Sigrist (2006), where $\rho_0 = 917\,\mathrm{kg/m}^3$ is the density of ice. This illustrates the potential of tensile slab fracture. In the weak
layer, minimum principal normal stresses (compression) normalized to their rapid-loading compressive strength $\sigma_c^- = 2.6\,\mathrm{kPa}$
according to Reiweger et al. (2015) are shown, illustrating the potential for weak-layer collapse. We choose principal stresses
for the visualization because they allow for the assessment of complex stress states by incorporating several stress components.
Please refer to Appendix D for the calculation of principal stresses from model outputs.

While the present model (Fig. 6, top panel) does not capture the highly localized stresses at the contact point between skier
and slab observed in the FEA model (Fig. 6, bottom panel), the overall stress fields are in excellent agreement. This is consistent
with Saint-Venant's principle, according to which the far-field effect of localized loads is independent of their asymptotic near-
field behavior. The same holds for the displacement field. While the concentrated load introduces a dent in the slab's top
surface, the overall deformations agree. With respect to the numerical reference, the present model renders displacement fields
and both weak-layer and slab stresses well. Moreover, we can confirm the model assumption of constant stresses through the
thickness of the weak layer.

Experimental validations are challenging since direct measurements of stresses are not possible and displacement measure-
ments require considerable experimental effort. The latter can be recorded using digital image correlation (DIC) as demon-
strated by Bergfeld et al. (2022). From their analysis, we use the DIC-recorded displacement field of the first $1.3\,\mathrm{m}$ of a
$3.0 \pm 0.1\,\mathrm{m}$ long flat-field propagation saw test (Fig. 7, bottom panel). The PST was performed on January 7, 2019, had a slab
thickness of $h = 46\,\mathrm{cm}$, a critical cut length of $a = 23 \pm 2\,\mathrm{cm}$, and the density profile shown in Fig. 7 (left panel) with a mean
slab density of $\bar{\rho} = 111 \pm 6\,\mathrm{kg/m}^3$. From the density we computed individual layer stiffnesses according to Eq. (20). Fig. 7





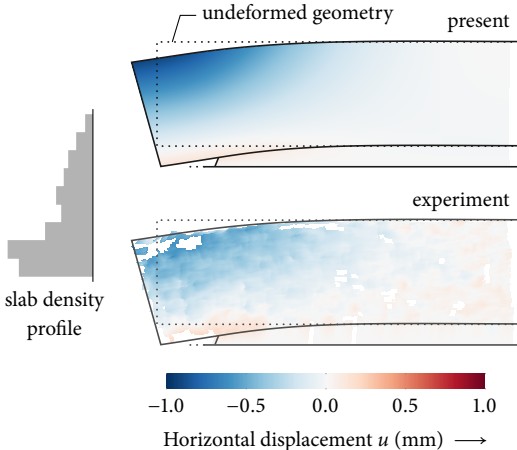

**Figure 7.** Horizontal displacement field of the first $1.3\,\mathrm{m}$ of a flat-field propagation saw test (PST) with an $a = 23\,\mathrm{cm}$ cut into the $t = 1\,\mathrm{cm}$ weak layer under a $h = 46\,\mathrm{cm}$ slab. Comparison of the present model (top) with full-field digital image correlation measurements (bottom). White patches indicate missing data points. Deformations are scaled by a factor of 100 and the weak-layer thickness by a factor of 10 for illustration.

compares both in-plane deformations of the snowpack (outlines) and the horizontal displacement fields (colorized overlay) obtained from the present model (top panel) and from DIC measurements (bottom panel). Deformations are scaled by a factor of 100, the weak-layer thickness by a factor of 10 for their visualization. In-plane slab and weak-layer deformations are in

very good agreement. This is evident in both the deformed contours and in the colorized displacement field overlay. Since displacements are $\mathcal{C}^1$-continuous across layer interfaces, the effect of layering is not directly visible in the displacement field. However, the slightly larger-than-expected tilt of the slab at its left end hints at a higher stiffness at the bottom of the slab and a compliant top section.

### 3.3    Weak-layer stresses and energy release rates

For all benchmark profiles illustrated in Fig. 5, weak-layer shear and normal stresses $(\tau, \sigma)$ obtained from the FEA model (dotted, light) and the present analytical solution (solid, dark) are compared in Fig. 8. We investigate a $38°$ inclined slope subjected to a concentrated force equivalent to the load of an $80\,\mathrm{kg}$ skier on an effective out-of-plane ski length of $1\,\mathrm{m}$. The finite element reference model has a horizontal length of $10\,\mathrm{m}$, of which the central $3\,\mathrm{m}$ are shown. The boundary conditions of the present model require the complementary solution (12) to vanish, representing an infinite extension of the system.

Kinks in the model solution originate from the loading discontinuity introduced by the concentrated skier force. They are a direct result of the plate-theory modeling approach. The agreement with the FEA reference solution is close for all types of investigated profiles and layering effects on weak-layer stress distributions are well captured. Only for profile ❏ C, the present solution slightly underestimates the normal stress peak directly below the skier. As Rosendahl and Weißgraeber (2020b) argue, this observation is inconsequential for weak-layer failure prediction. They discussed that accurately capturing size effects




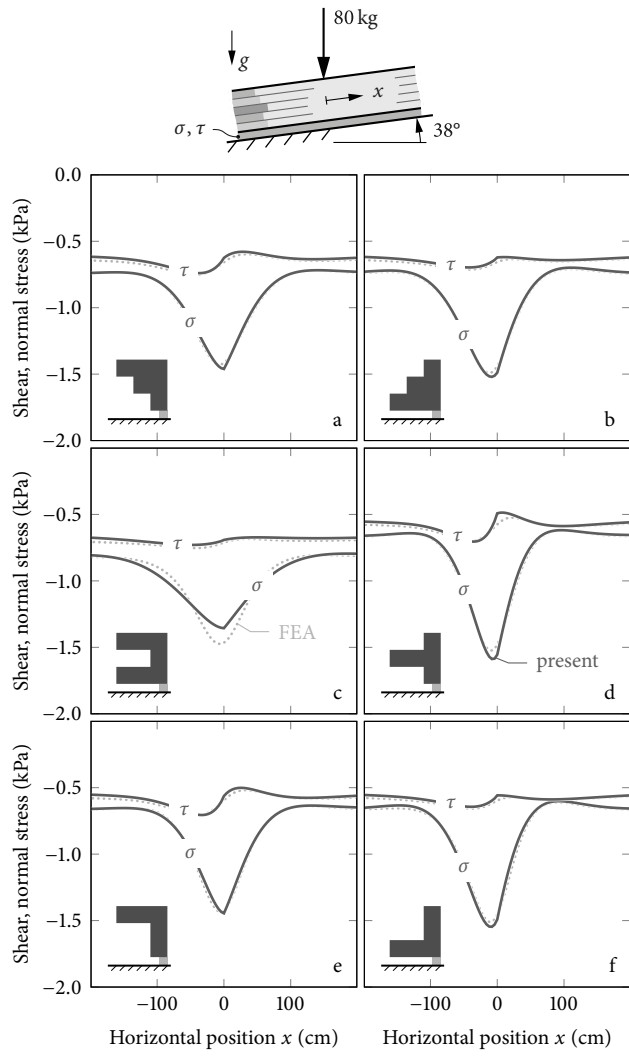

**Figure 8.** Weak-layer normal and shear stresses ($\sigma, \tau$) owing to combined skier and snowpack-weight loading for the benchmark profiles illustrated in Fig. 5. The present solution (solid, dark) only slightly underestimates the maximum normal stresses with respect to the FEA reference (dotted, light) in the case of profile ⊒ C. Material properties are given in Tables 1 and 2.

present in any structure, requires the evaluation of stresses in a certain distance from their peak (Neuber, 1936; Peterson, 1938; Waddoups et al., 1971; Sih, 1974; Leguillon, 2002; Weißgraeber et al., 2015; Rosendahl et al., 2019). Effects of bending stiffness (Fig. 8c vs. d) or bending–extension coupling (Fig. 8e vs. f) resulting from different layering orders, will be discussed in detail below.

A similar comparison of solutions for all profiles is given in Fig. 9, where total energy release rates (ERRs) of weak-layer
anticracks in 38° inclined PST experiments are shown. Here, both models consider free boundaries of the $1.2\,\mathrm{m}$ long PST block. The structure is loaded by the weight of the slab and saw-introduced cracks are modeled by removing all weak-layer


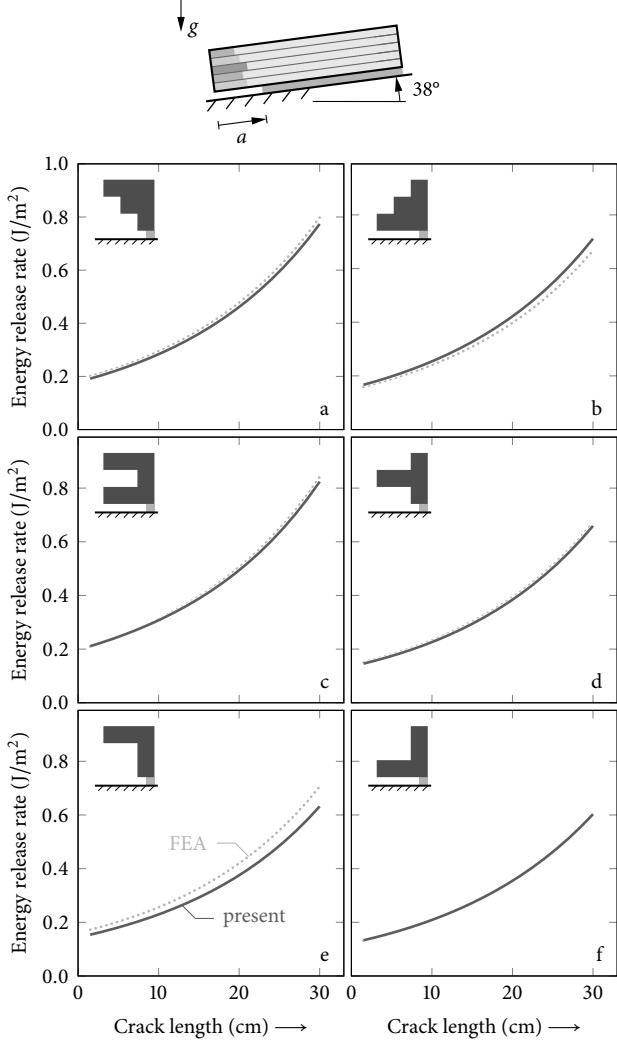

**Figure 9.** Total energy release rates of weak-layer anticracks in 38° inclined PST experiments of 120 cm length with the benchmark profiles illustrated in Fig. 5. The present solution (solid, dark) and FEA reference (dotted, light) are in good agreement. Material properties are given in Tables 1 and 2.

elements on the crack length $a$. This causes finite ERRs, even for very small cracks, because a finite amount of strain energy is removed from the system with these elements. The ERR of a sharp crack is expected to vanish in the limit of zero crack length ($\ll 1$ cm).

The principal behavior of the ERR as a function of crack length is unaffected by the choice of profile. However, the different resulting stiffness and deformation properties influence the magnitude of the energy release rate considerably. For instance, between cases A and B, we observe a difference of almost 10 % (Fig. 9).





**Table 3.** Slab extension, coupling, bending, and shear stiffnesses of the benchmark profiles. Comparison of $A_{11}, B_{11}, D_{11}$, and $A_{55}$ of the present model with $A_{11}^{\mathrm{eq}}, B_{11}^{\mathrm{eq}}, D_{11}^{\mathrm{eq}}$, and $A_{55}^{\mathrm{eq}}$ obtained from an equivalent isotropic slab according to Monti et al. (2015). Numbers in parentheses indicate the ratio of the modeled stiffness to the corresponding stiffness obtained from finite element analyses (visualized in Fig. 10).

| | | | | | | | | |
|---|---|---|---|---|---|---|---|---|
| present | $A_{11}$ ($10^4$ N/mm) | 1.65 (1.0) | 1.65 (1.0) | 2.47 (1.0) | 1.33 (1.0) | 1.33 (1.0) | 1.33 (1.0) | 1.15 (1.0) |
| | $B_{11}$ ($10^6$ N) | $-1.36$ (1.0) | 1.36 (1.0) | 0.00 (1.0) | 0.00 (1.0) | $-1.36$ (1.0) | 1.36 (1.0) | 0.00 (1.0) |
| | $D_{11}$ ($10^8$ Nmm) | 2.02 (1.0) | 2.02 (1.0) | 3.75 (1.0) | 0.34 (1.0) | 1.98 (1.0) | 1.98 (1.0) | 1.24 (1.0) |
| | $A_{55}$ ($10^3$ N/mm) | 6.44 (1.0) | 6.44 (1.0) | 9.63 (1.0) | 5.19 (1.0) | 5.19 (1.0) | 5.19 (1.0) | 4.32 (1.0) |
| Monti et al. | $A_{11}^{\mathrm{eq}}$ ($10^4$ N/mm) | 1.17 (0.7) | 1.17 (0.7) | 1.79 (0.7) | 0.72 (0.5) | 0.72 (0.5) | 0.72 (0.5) | 1.15 (1.0) |
| | $B_{11}^{\mathrm{eq}}$ ($10^6$ N) | 0.00 (0.0) | 0.00 (0.0) | 0.00 (1.0) | 0.00 (1.0) | 0.00 (0.0) | 0.00 (0.0) | 0.00 (1.0) |
| | $D_{11}^{\mathrm{eq}}$ ($10^8$ Nmm) | 1.26 (0.6) | 1.26 (0.6) | 1.93 (0.5) | 0.78 (2.3) | 0.78 (0.4) | 0.78 (0.4) | 1.24 (1.0) |
| | $A_{55}^{\mathrm{eq}}$ ($10^3$ N/mm) | 4.38 (0.7) | 4.38 (0.7) | 6.71 (0.7) | 2.69 (0.5) | 2.69 (0.5) | 2.69 (0.5) | 4.32 (1.0) |

## 4 Results

In the following, we use the above model to conduct parametric studies in order to investigate key mechanisms that may or may not lead to the release of slab avalanches. Among these are bridging or the effect of layer ordering. Unless specified otherwise, we used the material parameters given in Tables 1 and 2.

### 4.1 Stiffnesses of layered slabs

The mechanical behavior of the slab is governed by its stiffnesses. A layered system may have different stiffnesses with respect to extension, shear, or bending. Hence, we distinguish the extensional stiffness $A_{11}$, the bending–extension coupling stiffness $B_{11}$, the bending stiffness $D_{11}$, and the shear stiffness $A_{55}$. They are obtained from integration of the individual layer stiffnesses as specified in Eqs. (8a) to (8d). The ordering of layers influences each stiffness differently. That is, the simple homogenization of layered continua in the form of a single homogeneous equivalent layer is insufficient. Focusing on shear stress only, Monti et al. (2015) proposed a concept of equivalent layers to allow for the use of Boussinesq's solution for an isotropic elastic half-plane. They followed concepts developed in order to describe the surface deformation of layered systems in normal direction (De Barros, 1966). Using the equivalent Young's modulus $E_{\mathrm{eq}}$ introduced by Monti et al. (2015), the




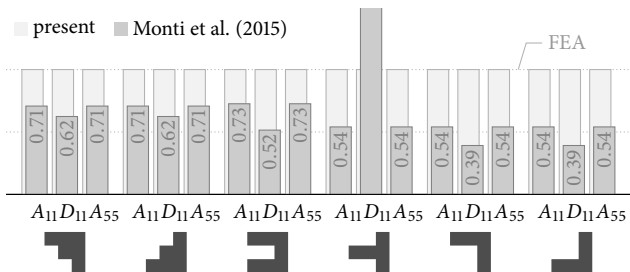

**Figure 10.** Slab extension, bending, and shear stiffnesses $A_{11}$ (N/mm), $D_{11}$ (Nmm), and $A_{55}$ (N/mm) of the present model and the equivalent isotropic slab approach by Monti et al. (2015) normalized to the finite element analysis (FEA) reference stiffness. The bending–extension coupling stiffness $B_{11}$ (N) is not shown because it is always zero in the model of Monti et al. (2015) and agrees exactly between reference and present model, see Table 3.

stiffnesses of a homogenized slab read

$$A_{11}^{\text{eq}} = \frac{E_{\text{eq}} h}{1 - \nu^2}, \tag{23a}$$

$$B_{11}^{\text{eq}} = 0, \tag{23b}$$

$$D_{11}^{\text{eq}} = \frac{E_{\text{eq}} h^3}{12 \left(1 - \nu^2\right)}, \tag{23c}$$

$$A_{55}^{\text{eq}} = \frac{E_{\text{eq}} h}{2 (1 + \nu)}. \tag{23d}$$

Table 3 and Fig. 10 compare stiffnesses computed with the present concept of laminate mechanics, Eqs. (8a) to (8d), with these stiffnesses of an equivalent homogeneous slab computed with properties obtained from the equivalence concept, Eqs. (23a) to (23d). Both concepts are benchmarked against the stiffnesses computed using finite element analyses. Here, the corresponding stiffnesses are obtained from the force response of unit extension and bending deformations. While Eqs. (8a) to (8d)

reproduce the reference stiffnesses exactly, the equivalent layer approach systematically underestimates the extensional, the bending, and the shear stiffnesses and cannot account for bending–extension couplings.

### 4.2 Effect of layering

To study the effect of layering we look at the deformations within a PST of $250\,\text{cm}$ length with a $50\,\text{cm}$ cut (20% of the PST length). The symmetric configuration of profile ⌐ C is studied as well as the profiles ◥ A and ◢ B with typical layerings. The

results are shown in Fig. 11. Here, the unsupported length of the slab is illustrated by a shaded background. The longitudinal displacement of the midplane $u_0$ and at the interface between the slab and the weak layer $\bar{u}$ show pronounced effects around the crack tip that induces slab bending. The midplane deformation of the symmetric profile ⌐ C is practically unaffected by this bending since its bending–extension stiffness $B_{11}$ is zero (Table 3). That is, bending and extension are only coupled through the weak layer but not through the slab itself. The near-constant midplane displacements originate from the 38° inclination. For

the asymmetric profiles, the effect of slab bending depends on the stiffness distribution. The stiff bottom layer of profile ◢ B

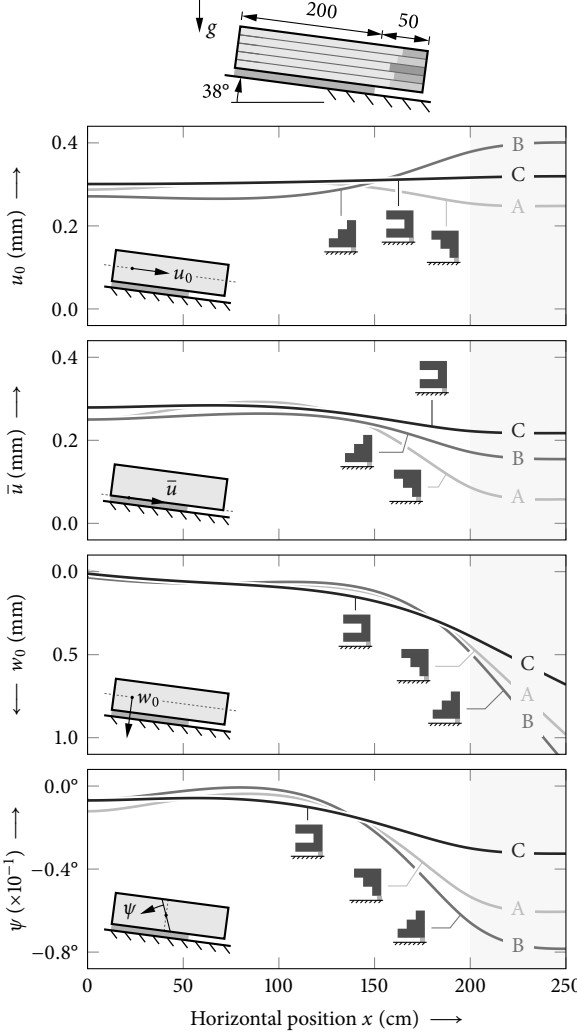

**Figure 11.** Deformations along the length of a PST with a cut at $x = 200\,\mathrm{cm}$ (cut length $50\,\mathrm{cm}$) illustrated by the shaded background. Comparison of three snow profiles. The longitudinal displacement of the midplane of the slab $u_0$ and at the interface between slab and weak layer $\bar{u}$, the deflection $w_0$, and the cross-section rotation $\psi$ are shown.

increases midplane displacements when the slab bends down on towards the right end of the PST. The opposite is observed for profile ◪ A with a stiff top layer. Here, the midplane displacements are reduced owing to crack-induced slab bending. The effect can be attributed to the different signs of the bending–extension stiffnesses $B_{11}$ of profiles ◪ A and ◪ B (Table 3). Constant longitudinal displacements at the interface between slab and weak layer $\bar{u}$ are reduced by slab bending for all profiles.

Profile ◪ C has the largest bending stiffness $D_{11}$ (Table 3). Hence, its reduction of $\bar{u}$ is smallest. Again, the stiff top layer of profile ◪ A causes a strong reduction of axial displacements. Deflections $w_0$ are downward positive (compression of the weak





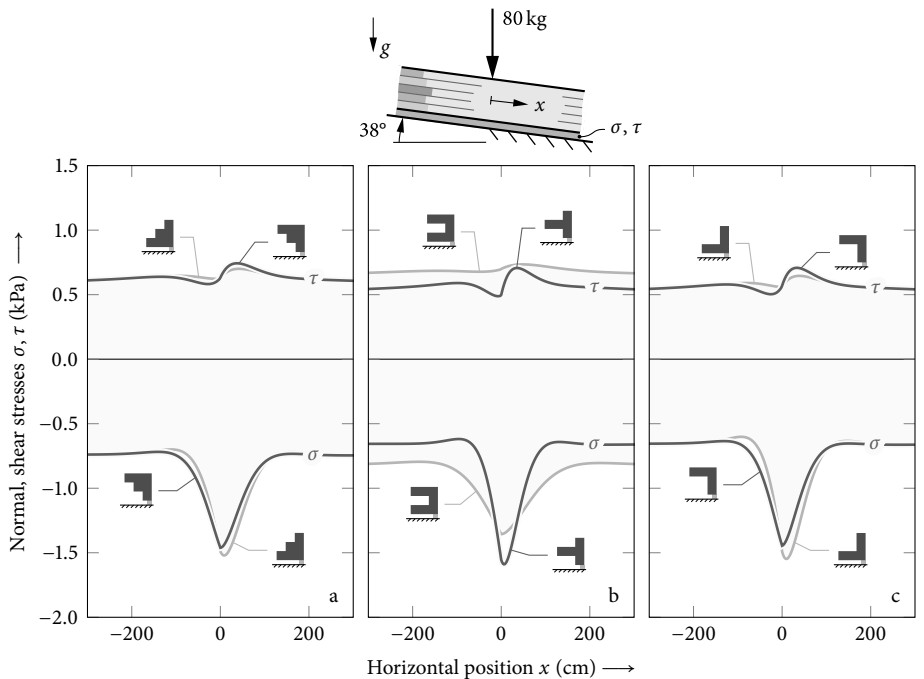

**Figure 12.** Comparison of shear and normal stresses in the weak layer of inclined skier-loaded layered snowpacks. The central $0.6\,\mathrm{m}$ section of an infinite slab is shown.

layer) along the complete PST and increase towards the cut end. Again, profile ⊐ C has the largest bending stiffness and, hence, exhibits the smallest deflections. Soft top layers (profile ◢ B) cause the largest deflections. Cross-section rotations $\psi$ are close to zero in the longitudinal center of the PST and increase towards the free ends of the PST, where the negative sign indicates

down bending. Similar arguments as for $w_0$ hold.

The effect of layering on the stresses in the weak layer is illustrated in Fig. 12. It shows shear and normal stresses in the weak layer below a skier-loaded slab, each panel for two of the considered profiles. Since the profiles ◥ A and ◢ B and profiles ◥ E and ◢ F have the same mean densities, their stress levels outside the skier's influence zone are the same. Profiles ⊐ C and ◣ D have a different mean densities and, hence, the stresses induced by the slab weight outside the skier's influence are

different. Here, constant loading leads to constant slab deformations and, hence, to constant weak layer stresses. Both shear and normal stresses show pronounced stress peaks close to the skier load point. As discussed above (Table 3), owing to their layering, profiles ⊐ C and ◣ D differ significantly in their bending stiffnesses (factor of 11) while the extensional stiffness is only doubled. In particular the smaller bending stiffness of profile ◣ D leads to localized stresses below the skier with higher maximum values but narrower influence zones (Fig. 12b). In the comparison of profiles ◥ A and ◢ B (Fig. 12a) and profiles

◥ E and ◣ F (Fig. 12c), we observe that profiles with increasing top-to-bottom stiffness exhibit slightly stronger weak-layer normal stress concentrations but weaker shear stress concentrations compared to their counterparts with reverse layering order.





In Fig. 13, the energy release rates of cuts introduced in PST experiments are shown as a function of crack length. For each pair of two profiles (A–B, C–D, E–F), the total differential energy release rate is shown. All curves show the expected monotonic increase of the energy release rate with increasing crack length. However, magnitudes and the progression towards

higher crack lengths strongly depend on the layering. The comparisons of profiles ⌐A vs. ⌐B (Fig. 13a) and ⌐E vs. ⌐F (Fig. 13c) illustrate that even with same extensional and bending stiffnesses, the order of layers has a significant impact on the energy released during crack growth. As observed in Fig. 12, profiles with increasing top-to-bottom stiffness are more critical with respect to the weak layer's structural integrity. The energy release rate depends on both the compliance of the snowpack and on the overall loading. That is, layers of higher density represent increased weight loads but since the Young modulus

increases with increasing stiffness, deformations of the slab and energy release rates may decrease. This is evident in Fig. 13b. Here, profile ⌐C is heavier than profile ⌐D. However, owing to its increased stiffness, its energy release rate is smaller.

### 4.3 Bridging

The distribution of a localized external load over a certain area of the weak layer (bridging) depends on the stiffness of the slab. To study this important effect, Fig. 14 shows skier-induced weak-layer stresses below a slab with profile ⌐F in its original and a

modified configuration. For the modification, the thicknesses of all layers of the original profile given in Table 1 are halved. The reduced weight ($\rho \propto h$) of the thinner slab leads to smaller overall stresses. However, its reduced stiffness ($A_{11} \propto h, D_{11} \propto h^3$) yields more pronounced stress peaks. In the case of normal stresses, peak compressive stresses below the thinner slab even exceed the ones of the original configuration. For shear stresses, the sharper stress peak does not outweigh the reduced slab weight.

While the effect of bridging on weak-layer stresses through the distribution of concentrated loads is somewhat intuitive, it can be observed for the energy release rate of weak-layer anticracks, too. Let us demonstrate this by investigating total thickness changes of layered slabs in PST experiments. Figure 15a shows the energy release rates of a cut of $a = 30\,\mathrm{cm}$ length in a 2.5 m

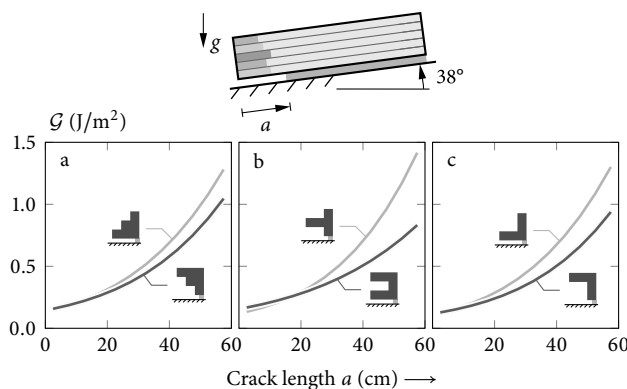

**Figure 13.** Comparison of total differential energy release rates $\mathcal{G}$ of cracks of length $a$ in a 2.5 m long PST between the considered benchmark profiles.


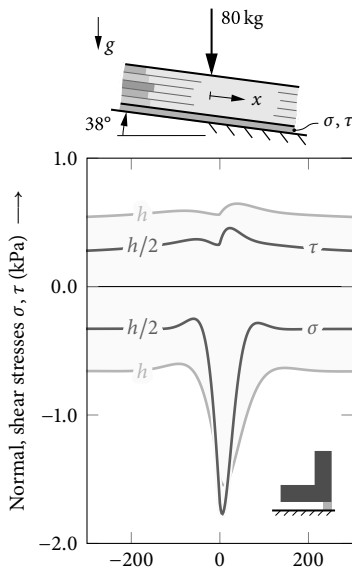

**Figure 14.** Effect of changes of the slab thickness $h$ on shear and normal stress in the weak layer under skier loading, shown for profile F.

propagation saw test. Energy release rates are shown as functions of the total slab thickness for three different profiles (A, C, F). They increase with increased slab thickness, mainly because the energy release rate is proportional to the square

of the total load. At large slab thicknesses ($h > 70\,\mathrm{cm}$), the heaviest profile C shows the highest energy release rates and the lightest profile F the smallest. For small slab thicknesses ($h < 70\,\mathrm{cm}$), the opposite is observed. This can be attributed to the changing bending stiffness of the slab. In order to isolate the influence of slab stiffness, Fig. 15b shows the energy release rate normalized by the square of the slab weight $\sum \rho_i h_i$. Since flat PSTs are dominated by the slab's bending stiffness, which again has a cubic dependence on the slab thickness ($D_{11} \propto h^3$), we observe a sharp decrease of the weight-normalized energy

release rates with increasing slab thickness, i.e., increasing slab bending stiffness. Hence, profile C with the highest bending stiffness (Table 3) has the lowest normalized energy release rate and profile F with the highest compliance (Table 3) exhibits the highest normalized energy release rate.

### 4.4 Effect of slope angle

The slope angle has a particular effect on the mode I/II mixety (compression and shear) of energy release rates in propagation

saw tests. Consider the $2.5\,\mathrm{m}$ PST with $a = 50\,\mathrm{cm}$ cuts between inclinations $-90° \leq \varphi \leq 90°$ shown in Fig. 16. All PSTs are cut from the right-hand side such that negative slope angles ($\varphi < 0$) correspond to upslope cuts and positive slope angles ($\varphi > 0$) to downslope cuts. Profiles B, C, D, and the homogeneous case H are shown. With increasing inclinations (both positive and negative) shear stresses and deformations increase. This increases the mode II energy release rate and, hence, the mixed mode ratios $\mathcal{G}_{II}/\mathcal{G}_I$. However, common effect for all profiles are considerably larger mixed mode ratios $\mathcal{G}_{II}/\mathcal{G}_I$ for

downslope cuts ($\varphi > 0$). While mode II energy release rates reach the magnitude of their mode I counterparts $\mathcal{G}_{II}/\mathcal{G}_I \approx 1$ at



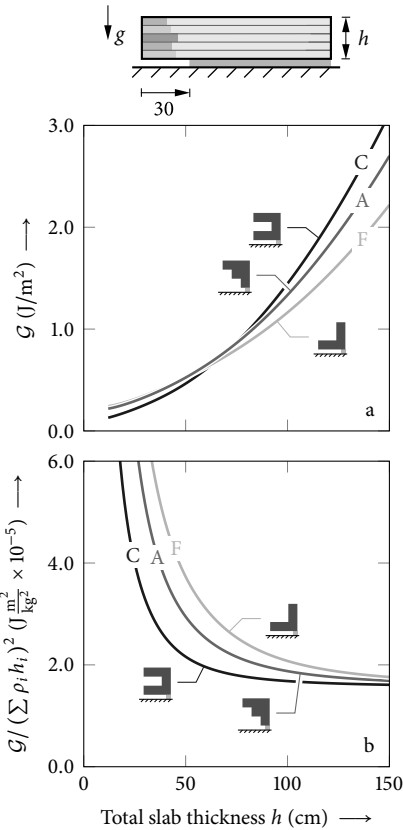

**Figure 15.** Bridging effect on the energy release rate in flat PST experiments. a) Total differential energy release rate $\mathcal{G}$ for profiles ⊐ C, ◄ D, and ⊐ E. b) Energy release rate $\mathcal{G}$ normalized with respect to the square of the total slab weight $\sum \rho_i h_i$.

$\varphi \approx +45°$, this magnitude is first reached at $\varphi \approx -70°$ for upslope cuts. The effect can be amplified by the slab's layering. While the homogeneous profile ▮ H and profile ⊐ C produce notable mode Ⅱ contributions in upslope cuts, profile ◄ D makes mode Ⅱ energy release rates almost inaccessible with upslope PSTs.

The effect originates from the competition of different shear stress contributions. Unsupported sections of the slab cause transverse shear forces at the crack tip that induce transverse shear stresses. The shear forces originate from the slab's gravitational dead load and, hence, induce shear stresses of the same sign regardless of slope angle. Then again, horizontal slab movements on inclined slopes induce lateral shear stresses that change their sign with slope angle. At the upslope ends of PSTs, both shear stresses have the same sign and cause considerable contributions to the mode Ⅱ energy release rate for downslope cuts. At the downslope end of PSTs, the shear stresses have opposite signs inducing small mode Ⅱ contributions for upslope

cuts.

This has important implications for field tests. If pure mode I energy release rates are of interest, upslope cuts are relatively robust against mode Ⅱ influences. However, if mode Ⅱ contributions are of interest, downslope cuts are advised.



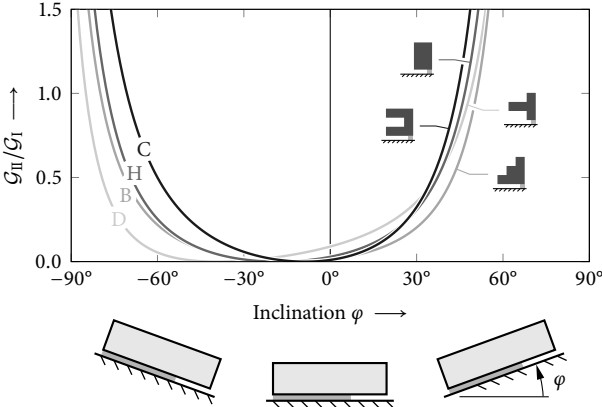

**Figure 16.** Effect of slope angle on the mode mixity of the energy release rates in propagation saw tests. Mode mixety is expressed as the ratio of mode II (shear) to mode I (collapse) energy release rate ($\mathcal{G}_{II}/\mathcal{G}_I$). PSTs are $2.5\,\mathrm{m}$ long and cut $a = 50\,\mathrm{cm}$ from the right.

## 4.5 Example of extended analyses

As discussed in Section 2.4, the model covers complex cases with multiple external loads and several interacting cracks. An example is given in Fig. 17 where an inclined snowpack with profile ◢ B is loaded by two skiers in the vicinity of a weak-layer crack. For this analysis, five segments connected through transmission conditions were introduced to account for the discontinuities of two external loads and the crack. Figure 17a shows the obtained slab displacements and the rotations of slab cross sections. Both skiers locally increase deformations and interact, in particular with respect to deflections $w_0$, owing to their proximity. The deformations of the layered slab above the crack of $100\,\mathrm{cm}$ length are even larger, yet, much smaller than the weak-layer thickness of $20\,\mathrm{mm}$. Figure 17b shows the corresponding weak-layer shear and normal stresses. Again, the interaction of both loads, in particular in terms of normal stresses, is observed. Without load interaction, stresses would drop to the level of stresses induced by the slab weight alone in between the skiers. The effect is connected to bridging because the area across which individual loads are distributed depends on the snowpack's stiffness.

## 5 Discussion

The proposed model uses the established concepts of laminate mechanics to assess the problem of layered slabs resting on weak layers. Heierli (2008) and Rosendahl and Weißgraeber (2020a) have shown that beam-type solutions can provide accurate representations of the mechanical response of homogeneous snowpacks loaded by gravity and localized loads. Analyses of layered snowpacks have only been performed with numerical models (Schweizer, 1993; Habermann et al., 2008) or with approximate solutions of limited generality (Monti et al., 2015). The validation in Section 3 shows that the present model provides an accurate closed-form analytical solution for layered slabs on a weak layer loaded by their own weight and external (point) loads. The comparison to the numerical reference solution demonstrates a high accuracy of the solution in terms of

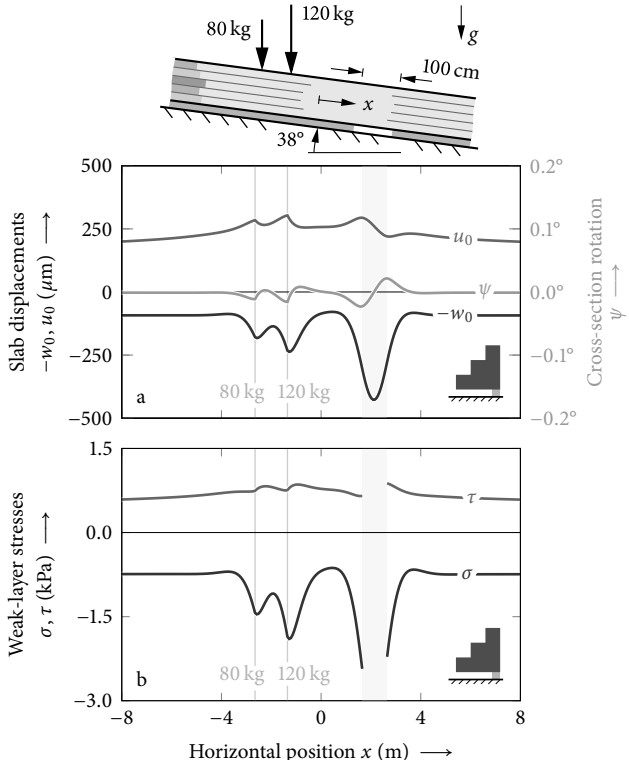

**Figure 17.** Example of a complex configuration with two skier loads on profile ◢ B in the vicinity of a 100 cm weak-layer crack. Note that positive deflections $w_0$ point in the physical downward direction. Here, we show $-w_0$ to maintain the intuitive downward direction of a positive $w_0$ when displayed on the same abscissa as $u_0$.

displacements, stresses, and also energy release rates of anticracks within the weak layer. The latter is obtained by using the analysis approaches developed for so-called weak interfaces exhibiting high elastic contrasts (Fraisse and Schmit, 1993; Lenci, 2001).

The anisotropic mechanical response of the slab is described by the stiffnesses of laminate mechanics. The extensional stiffness $A_{11}$ and the shear stiffness $A_{55}$ are linear with respect to the thickness of the individual layers within the slab and do not depend on the ordering. The bending–extension coupling stiffness $B_{11}$ is zero for symmetric laminates and scales both with the square of the individual layer thickness and linearly with z-distance to the coordinate origin. Hence, it depends on the order of layers. This is even more pronounced for the bending stiffness $D_{11}$ that depends on the power of three of the

layer thicknesses and on the square of the distance to midplane. That is, both stiffnesses account for the complex mechanical behavior of a layered structure while accounting for layer ordering effects. Table 3 shows that within the considered examples, decisive differences between the stiffnesses of different profiles can occur. The profile pairs ◤ A, ◢ B and ◤ E, ◢ F each have the same extensional and bending stiffnesses, $A_{11}$ and $D_{11}$, respectively, and only the sign of the bending–extension stiffness $B_{11}$ differs. Profiles ◳ C and ◰ D exhibit a strong layering effect. In the equivalent-layer concept (Monti et al., 2015), the layer





moduli are homogenized into one equivalent Young's modulus of the slab. To use models for homogeneous elastic half-spaces (e.g., Föhn, 1987), this system of slab and weak layer is then replaced with a single layer with the Young's modulus of the weak layer and the slab thickness is scaled to account for this. Of course, such a homogenization must work for extension deformation as well as bending deformation. However, Table 3 and Fig. 10 show that using this concept does not yield correct stiffness properties of the slab. As pointed out by Monti et al. (2015), the equivalence layer concept does not account for the
order of the layers. Hence, the significant ordering effects of the considered profiles cannot be not accounted for.

   Birkeland et al. (2014) address the role of the slab on the crack propagation. They changed the slab by introducing cuts normal to the surface that significantly reduce the thickness locally. As shown in Fig. 15, when normalized for different profile weights, the reduced bending stiffness leads to much lower energy release rates that may not suffice for crack propagation. In a PST experiment, the weight of the slab is the only load and is constant along the weak layer. In a skier-loaded snowpack, the
local loading of the skier leads to a locally increased energy release rate in the vicinity of the skier. With low bending stiffness, this energy release rate attains locally high values but then rapidly decreases to energy release rates originating from the slab's weight only. With higher bending stiffnesses, the influenced domain of a localized loading (e.g., a skier) is larger while the magnitude of the effect decreases.

   The deformations of the slab (Fig. 11) show the resulting effect of the layering. This is pronounced as the longitudinal
deformation at the interface of the slab and the weak layer $\bar{u}$ depends strongly on the beam rotation $\psi$. That is, with increased bending stiffness of a slab, the longitudinal deformations at the weak layer will also be smaller leading to reduced shear loading of the weak layer. The analysis of the stresses in the weak layer (Fig. 12) shows that the layering and the order of the layers control weak layer stresses and the effective bridging length (Schweizer and Camponovo, 2001a). In particular, the stress peaks below the localized loading of the skier will change with bridging. For stiffer slabs, a wider area below the skier is loaded while
the maximum stresses decrease. Besides the stress loading in the weak layer, the energy released during crack initiation and growth controls avalanche release. The energy release rate, too, shows a pronounced effect of the stiffness of the slab and the ordering of the layers (Fig. 13). Slabs with high stiffness layers adjacent to the weak layer lead to higher energy release rates (in the considered PST configuration). The present results agree with the findings by Schweizer and Jamieson (2003), van Herwijnen and Jamieson (2007), and Thumlert and Jamieson (2014) that identified an increase of snowpack stability with
increased bridging. Moreover, the results of the current model on the energy release rate of layered slabs can explain why failure propagation may be accentuated by stiff slabs, also reported by van Herwijnen and Jamieson (2007).

   In the studies by Schweizer and Jamieson (2003) and Thumlert and Jamieson (2014), a bridging index (BI) is introduced and applied to the analysis of snowpack stability. The bridging index accounts for the hand hardness index and the thickness of each layer. We propose to use the bending stiffness $D_{11}$ to characterize the bridging of a snowpack configuration. Then, the
ordering of the layers and the nonlinear contribution of the thickness to the bending behavior is considered. By restricting the consideration to this single property, effects such as shear deformation, bending–extension coupling, or weak layer deformation are not considered but it will provide a good first indication of the bridging. For a full analysis, the use of a comprehensive and efficient model like the present one is advised.





The effect of the stiffness is also studied at hand of profiles, in which the layer order remains the same but each layer

thickness is changed by the same factor (Fig. 14). With half the thickness of each layer, the total bending stiffness is reduced by

a factor of 8. Hence, the bridging area is reduced and the maximum peak stress increases although the general stress level in the

weak layer has decreased due to the lower total weight of the thinner layered slab. For energy release rates in PST experiments,

weight loading dominates and heavier profiles (⌐ C > ⌐ A > ⌐ F) feature higher energy release rates (Fig. 15a). Only when

normalizing for the slab weight, an increased bending stiffness (⌐ C > ⌐ A > ⌐ F) reduces the energy release (Fig. 15b).

Investigating the effect of the slope angle on energy release rates of PST experiments (Fig. 16) offers intriguing views of the

behavior of PSTs and its experimental variants. The slab above the cut is subject to two sources of shear loading: i) transverse

shear deformation from the shear force of the weight of the overhanging slab and ii) lateral shear loading of the tangential

component $q_t$ of the gravitational load. On a flat slope, the latter vanishes. On inclines, its sign changes with negative and

positive slope angles. The former has the same sign regardless of positive or negative inclination. Hence, shear contributions

to the energy release rate are superimposed either additively or subtractively depending on the sign of the slope angle. Our

results show that for upslope cuts, mode Ⅱ plays a much smaller role than for downslope slope cuts. This has a direct effect

on the mode Ⅱ energy release rate and constitutes a significant difference between the two possible cut directions. Sigrist and

Schweizer (2007), who were able to obtain relatively large contributions from shear deformations in their PST experiments,

used downslope cuts. Whether this was done for the purpose of obtaining large mode Ⅱ contributions or coincidence is not

reported but consistent with the present results. The findings may be used to develop PST procedures specifically designed to

study mode I and mode Ⅱ separately. Previously, some variations of PST experiments have been proposed in literature (e.g.,

Birkeland et al., 2019).

Even with increasing number of comprehensive numerical models, closed-form analytical models are highly relevant. As

pointed out in the broad review by Morin et al. (2020), there is still a large need for an improved understanding of snow physics

and for models that can assess snowpack stability. Especially for the use in model chains, extensive parametric studies, or in

optimizations, a very high computational efficiency is very important. Within this work we have performed a total number

of 6789 different analyses in the considered non-exhaustive parametric studies. This alone highlights the necessity of highly

efficient, functional mechanical models. Moreover, in their simplistic structure, analytical models reveal fundamental physical

interrelationships and effects. The present model in particular uses only input parameter with clear physical meaning that can

be determined in relatively simple experiments. No numerical stabilization such as artificial viscosity or tuning parameters for

complex constitutive laws that are not directly accessible in experiments are used or required.

Closed-form models as the present one are based on fundamental mechanisms and provide a window into the "how and why"

of the mechanics of dry-snow slab avalanche release. A similar model for homogeneous slabs (Rosendahl and Weißgraeber,

2020a) has been used by Bergfeld et al. (2021) to identify the Young's modulus of a slab by means of digital image correlation

of PST experiments. The authors observed that the model provided consistent results for the Young's modulus of slab and

weak layer, irrespective of experimentally recorded cut lengths. In contrast, using the expression of the system's elastic energy

provided by Heierli et al. (2008), as proposed by van Herwijnen et al. (2016), showed a significant dependence on the cut

length and led to inconsistent results. This can be attributed to the negligence of weak-layer elasticity by Heierli et al. (2008)




and demonstrates the importance of considering the principal features of a physical problem. In the case of slab avalanche
release, we view the mechanics of the layered slab and the weak layer as crucial.

For the proposed model, the computational effort does not change with domain size or number of considered layers. Computing the eigenvalues of the system Matrix $K$ of the governing ODE (11) represents the main computational effort. This is independent of the number of segments or layers, and only needs to be done once for any set of boundary conditions, load cases, and slope angles. Each segment adds six free coefficients, i.e., six degrees of freedom to the linear system of equations
of Eq. (18). This has virtually no impact on the computation effort even with 20 segments. In this case, timing 1000 stress evaluations yields a mean run time of $0.7\,\mathrm{ms}$ per analysis on a single 2.4 GHz Intel i9 Core.

The model does not account for contact of the slab with base layers or the remains of a collapsed weak layer. For long weak-layer cracks, the corresponding normal deformations may become too large to be rendered correctly in the present model. A corresponding extension of the present model is work in progress and will allow for the analysis of sustained anticrack growth.

## 6 Conclusions

The present work presents a closed-form analytical model for the mechanical response of layered slab resting on compliant weak layers:

1. It is applicable to slopes loaded by one or multiple skiers and propagation saw tests.

2. The model provides anisotropic slab stiffnesses, slab displacement fields, weak-layer stresses, and energy release rates
of cracks in the weak layer that are in excellent agreement with finite element reference solutions.

3. Its implementation is highly efficient, allows for real-time applications, and for the consideration of arbitrary system sizes and an arbitrary number of layers.

4. In an analysis of bridging, we reveal significant effects of slab weight, stiffness, and layering on weak-layer stresses and energy release rates.

5. Based on an investigation pf inclined propagation saw tests, we recommend upslope cut PSTs for the analyses for mode I energy release rates and downslope cut PSTs for mode II analyses.

**Appendix A: Derivation of the governing equations for a layered slab supported by an elastic foundation**

With the first derivative of the constitutive equation of the normal force (7a)′ inserted into the equilibrium of horizontal forces (6a), we obtain

$$0 = A_{11}u_0''(x) + B_{11}\psi_0''(x) + \tau(x) + q_{\mathrm{t}}. \tag{A1}$$





Likewise, with the first derivative of the constitutive equation of the shear force (7b)$'$ and the vertical force equilibrium (6b), we have:

$$0 = \kappa A_{55}(w_0''(x) + \psi'(x)) + \sigma(x) + q_{\mathrm{n}}. \tag{A2}$$

The first derivative of the constitutive equation of the bending moment (7a)$'$ with the balance of moments (6c), yields

$$0 = B_{11}u_0''(x) + D_{11}\psi''(x) - \kappa A_{55}\left(w_0'(x) + \psi(x)\right)$$
$$+ \frac{h+t}{2}\tau(x) + z_{\mathrm{s}}q_{\mathrm{t}}. \tag{A3}$$

We then insert the definition of the shear stresses (5b) into Eq. (A1) to obtain

$$0 = A_{11}u_0''(x) - k_{\mathrm{t}}u_0(x) - k_{\mathrm{t}}\frac{t}{2}w_0'(x)$$
$$+ B_{11}\psi''(x) - k_{\mathrm{t}}\frac{h}{2}\psi(x) + q_{\mathrm{t}}. \tag{A4}$$

Inserting the normal stress definition (5a) into Eq. (A2), yields

$$0 = \kappa A_{55}w_0''(x) - k_{\mathrm{n}}w_0(x) + \kappa A_{55}\psi'(x) + q_{\mathrm{n}}, \tag{A5}$$

and, again, inserting the shear stress (5b) into Eq. (A3), yields

$$0 = B_{11}u_0''(x) - k_{\mathrm{t}}\frac{h+t}{2}u_0(x) + D_{11}\psi''(x)$$
$$+ \left(\frac{h+t}{2}\frac{t}{2}k_{\mathrm{t}} - \kappa A_{55}\right)w_0'(x)$$
$$- \left(\kappa A_{55} + \frac{h+t}{2}\frac{h}{2}k_{\mathrm{t}}\right)\psi(x) + z_{\mathrm{s}}q_{\mathrm{t}}. \tag{A6}$$

Equations (A4) to (A6) constitute a system of linear ordinary differential equations of second order with constant coefficients of the deformation variables $u(x)$, $w(x)$, $\psi(x)$ that describes the mechanical behavior of a layered beam on a weak layer.

Using the vector $\boldsymbol{z}(x)$ of all unknown functions (10), the ODE system can be written as a system of first-order for the form

$$\boldsymbol{A}\boldsymbol{z}'(x) + \boldsymbol{B}\boldsymbol{z}(x) + \boldsymbol{d} = 0, \tag{A7}$$

with the matrices

$$\boldsymbol{A} = \begin{bmatrix} 1 & 0 & 0 & 0 & 0 & 0 \\ 0 & A_{11} & 0 & 0 & 0 & B_{11} \\ 0 & 0 & 1 & 0 & 0 & 0 \\ 0 & 0 & 0 & \kappa A_{55} & 0 & 0 \\ 0 & 0 & 0 & 0 & 1 & 0 \\ 0 & B_{11} & 0 & 0 & 0 & D_{11} \end{bmatrix}, \tag{A8}$$





and

$$
\boldsymbol{B} = \begin{bmatrix}
0 & -1 & 0 & 0 & 0 & 0 \\
-k_{\mathrm{t}} & 0 & 0 & k_{\mathrm{t}}\frac{t}{2} & -k_{\mathrm{t}}\frac{h}{2} & 0 \\
0 & 0 & 0 & -1 & 0 & 0 \\
0 & 0 & -k_{\mathrm{n}} & 0 & 0 & kA_{55} \\
0 & 0 & 0 & 0 & 0 & -1 \\
-\frac{h+t}{2}k_{\mathrm{t}} & 0 & 0 & B_{64} & B_{65} & 0
\end{bmatrix}, \tag{A9}
$$

where

$$
B_{64} = k_{\mathrm{t}}\frac{h+t}{4}t - \kappa A_{55}, \text{ and } B_{65} = -k_{\mathrm{t}}\frac{h+t}{4}h - \kappa A_{55},
$$

and the vector

$$
\boldsymbol{d} = \begin{bmatrix} 0 & q_{\mathrm{t}} & 0 & q_{\mathrm{n}} & 0 & z_{\mathrm{s}}q_{\mathrm{t}} \end{bmatrix}^{\mathsf{T}}. \tag{A10}
$$

The system (A7) can be rearranged into the form

$$
\boldsymbol{z}'(x) = \boldsymbol{K}\boldsymbol{z}(x) + \boldsymbol{q}, \tag{A11}
$$

where

$$\boldsymbol{K} = -\boldsymbol{A}^{-1}\boldsymbol{B}, \tag{A12}$$

$$\boldsymbol{q} = -\boldsymbol{A}^{-1}\boldsymbol{d}. \tag{A13}$$

## Appendix B: Derivation of the governing equations of an unsupported layered slab

Without elastic foundation, the equilibrium conditions (6a) and (6b) reduce to

$$
0 = \frac{\mathrm{d}N(x)}{\mathrm{d}x} + q_{\mathrm{t}}, \tag{B1}
$$

$$0 = \frac{\mathrm{d}V(x)}{\mathrm{d}x} + q_{\mathrm{n}}, \tag{B2}$$

$$
0 = \frac{\mathrm{d}M(x)}{\mathrm{d}x} - V(x) + z_{\mathrm{s}}q_{\mathrm{t}}. \tag{B3}
$$

By adding and subtracting $\pm D_{11}w_0''(x)$ to the constitutive equation of the bending moment (7a) and using the first derivative of the constitutive equation of the shear force (7b)′, we obtain

$$
M(x) = B_{11}u_0'(x) + \frac{D_{11}}{\kappa A_{55}}V'(x) - D_{11}w_0''(x). \tag{B4}
$$





Differentiating twice and using the first derivatives of the equilibrium conditions, (B2)′ and (B3)′, yields

$$M''(x) = V'(x) = -q_\mathrm{n} = B_{11} u_0'''(x) - D_{11} w_0''''(x). \tag{B5}$$

Adding and subtracting $\pm B_{11} w_0''$ to the constitutive equation of the normal force (7a) and using the constitutive equation of the shear force (7b), gives

$$N(x) = A_{11} u_0'(x) + \frac{B_{11}}{\kappa A_{55}} V'(x) - B_{11} w_0''(x). \tag{B6}$$

Differentiating this equation and, again, using the derivatives of the equilibrium conditions, (B1)′ and (B2)′, yields

$$N'(x) = -q_\mathrm{t} = A_{11} u_0''(x) - B_{11} w_0'''(x). \tag{B7}$$

Solving the derivative of this equation for $u_0'''(x)$ and inserting it into Eq. (B5), yields an ordinary differential equation of fourth order for the vertical displacement

$$w_0''''(x) = -\left( \frac{B_{11}^2}{A_{11}} - D_{11} \right) q_\mathrm{n}. \tag{B8}$$

It can be solved readily by direct integration

$$w_0(x) = C_1 + c_2 x + c_3 x^2 + c_4 x^3 - \left( \frac{B_{11}^2}{A_{11}} - D_{11} \right) q_\mathrm{n} x^4. \tag{B9}$$

Solving Eq. (B7) for $u_0''(x)$, integrating twice and inserting the third derivative of the general solution for $w_0(x)$ (B9)′, yields the general solution for the tangential displacement of unsupported beams

$$u_0(x) = c_5 + c_6 x + \frac{(6 B_{11} c_4 - q_\mathrm{t})}{2 A_{11}} x^2$$

$$- \frac{B_{11} q_\mathrm{n}}{6 \left( B_{11}^2 - A_{11} D_{11} \right)} q_\mathrm{n} x^3. \tag{B10}$$

To obtain a solution of the cross-section rotation $\psi(x)$, we take the derivative of the constitutive equation for the bending moment (7a)′ and insert it together with the constitutive equation of the shear force (7b) into the equilibrium of moments (B3). Solving this for $\psi(x)$ yields

$$\psi(x) = \frac{1}{\kappa A_{55}} \left( B_{11} u_0''(x) + D_{11} \psi''(x) + z_\mathrm{s} q_\mathrm{t} \right) - w_0'(x). \tag{B11}$$

Equation (B7) allows for eliminating $u_0''(x)$. In order to eliminate $\psi''(x)$, we insert the constitutive equation of the shear force (7b) into the second derivative of the vertical equilibrium (B2)″, which yields $\psi''(x) = -w_0'''(x)$ and we obtain

$$\psi(x) = \frac{B_{11}^2 - A_{11} D_{11}}{\kappa A_{55} A_{11}} w_0'''(x) - w_0'(x)$$

$$+ \left( z_\mathrm{s} - \frac{B_{11}}{A_{11}} \right) \frac{q_\mathrm{t}}{\kappa A_{55}}, \tag{B12}$$

which is fully defined through the solution for $w_0(x)$ (B9).





In order to assemble a global system of linear equations from boundary and transmission conditions between supported and unsupported beam segments, it is helpful to express the general solutions for both cases in the same form. For this purpose, we express vector of unknown functions (10) used for the solution of supported beam segments through the general solutions (B9), (B10) and (B12) for unsupported beam segments. This yields the matrix form $\boldsymbol{z}_\circ(x) = \boldsymbol{\mathcal{P}}(x)\,\boldsymbol{c}_\circ + \boldsymbol{p}(x)$, see Eq. (16), where $\boldsymbol{c}_\circ = \left[C_\circ^{(1)}, \ldots, C_\circ^{(6)}\right]^\mathsf{T}$ is the vector of unknown coefficients,


$$
\boldsymbol{\mathcal{P}}(x) =
\begin{bmatrix}
0 & 0 & 0 & 3\frac{B_{11}}{A_{11}}x^2 & 1 & x \\
0 & 0 & 0 & 6\frac{B_{11}}{A_{11}}x & 0 & 1 \\
1 & x & x^2 & x^3 & 0 & 0 \\
0 & 1 & 2x & 3x^2 & 0 & 0 \\
0 & -1 & -2x & \frac{6K_0}{A_{11}\kappa A_{55}} - 3x^2 & 0 & 0 \\
0 & 0 & -2 & -6x & 0 & 0
\end{bmatrix},
\tag{B13}
$$

and

$$
\boldsymbol{p}(x) =
\begin{bmatrix}
-\frac{q_t}{2A_{11}}x^2 - \frac{B_{11}}{6K_0}q_n x^3 \\
-\frac{q_t}{A_{11}}x - \frac{B_{11}}{2K_0}q_n x^2 \\
-\frac{A_{11}}{24K_0}q_n x^4 \\
-\frac{A_{11}}{6K_0}q_n x^3 \\
\frac{A_{11}}{6K_0}q_n x^3 + \left(z_s - \frac{B_{11}}{A_{11}}\right)\frac{q_t}{\kappa A_{55}} - \frac{q_n}{\kappa A_{55}}x \\
\frac{A_{11}}{2K_0}q_n x^2 - \frac{q_n}{\kappa A_{55}}
\end{bmatrix},
\tag{B14}
$$

with $K_0 = B_{11}^2 - A_{11}D_{11}$.

## Appendix C: Boundary and transmission conditions

Stability tests are typically conducted on finite volumes with free ends that require vanishing section forces and moments

$$
N = V = M = 0,
\tag{C1}
$$

as boundary conditions. Skier-induced loading is typically confined in a very small volume compared to the overall dimensions of the snowpack that extends over the entire slope. For the model, this corresponds to an unbounded domain where, all components of the solution converge to a constant at infinity. That is, at the boundaries, the complementary solution vector must 600 vanish

$$
\boldsymbol{z}_h = 0,
\tag{C2}
$$

which yields constant displacements $\boldsymbol{z}(x) = \boldsymbol{z}_p$, see Eq. (13).





At interfaces between two segments (e.g., change from supported to unsupported), $\mathcal{C}^0$-continuity of displacements and section forces is required and the transmission conditions read

$$\Delta u_0 = 0, \quad \Delta w_0 = 0, \quad \Delta\psi = 0,$$
$$\Delta N = 0, \quad \Delta V = 0, \quad \Delta M = 0, \tag{C3}$$

where the $\Delta$ operator indicates the difference between left and right segments, i.e., $\Delta y = y_{\mathrm{l}} - y_{\mathrm{r}}$. External concentrated forces (e.g., skiers) are introduced with their normal and tangential components $F_{\mathrm{n}}$ and $F_{\mathrm{t}}$ and with their resulting moment $M = -hF_{\mathrm{t}}/2$. This introduces discontinuities of the section forces that have to be accounted for in the form of the transmission

conditions

$$\Delta N = F_{\mathrm{t}}, \quad \Delta V = F_{\mathrm{n}}, \quad \Delta M = -\frac{h}{2}F_{\mathrm{t}}, \tag{C4}$$

where again, the $\Delta$ operator expresses the difference between left and right segments.

## Appendix D: Slab stress fields

The in-plane stresses $\sigma_x$, $\sigma_z$, and $\tau_{xz}$ within layers of the slab are obtained using the layers' constitutive equations and exploit-

ing the equilibrium conditions (Reddy, 2003). Using Hooke's law and the identities $\varepsilon_x(x,z) = u'(x,z) = u'_0(x) + z\psi'(x)$, the axial layer normal stresses can be expressed in terms of slab displacements in the form

$$\sigma_x(x,z) = \frac{E(z)}{1 - \nu(z)^2}\Big(u'_0(x) + z\psi'(x)\Big), \tag{D1}$$

where Young's modulus $E(z)$ and Poisson's ratio $\nu(z)$ are layerwise, i.e., piecewise, constant in $z$-direction. Integrating the local equilibrium condition

$$0 = \frac{\partial\sigma_x}{\partial x} + \frac{\partial\tau_{xy}}{\partial y} + \frac{\partial\tau_{xz}}{\partial z}, \tag{D2}$$

with respect to $z$, where derivatives with respect to $y$ vanish owing to the plane-strain assumption, yields the in-plane layer shear stress

$$\begin{aligned}
\tau_{xz}(x,z) &= -\int \sigma'_x(x,z)\,\mathrm{d}z \\
&= -\int \frac{E(z)}{1 - \nu(z)^2}\Big(u''_0(x) + z\psi''(x)\Big)\,\mathrm{d}z, \tag{D3}
\end{aligned}$$

The second-order derivatives are obtained from the left-hand side of Eq. (11) and integration with respect to $z$ is performed using the trapezoidal rule. Again, integrating the equilibrium condition

$$0 = \frac{\partial\tau_{xz}}{\partial x} + \frac{\partial\tau_{yz}}{\partial y} + \frac{\partial\sigma_z}{\partial z}, \tag{D4}$$



with respect to $z$ under the same assumptions, yields the interlayer normal stresses

$$\sigma_z(x,z) = -\int \tau'_{xz}(x,z)\,\mathrm{d}z. \tag{D5}$$

Here, differentiation is performed using difference quotients with consideration of discontinuities. Finally, maximum ($\sigma_{\mathrm{I}}$) and minimum ($\sigma_{\mathrm{III}}$) principal stresses are computed from

$$\sigma_{\mathrm{I,III}} = \frac{\sigma_x + \sigma_z}{2} \pm \sqrt{\left(\frac{\sigma_x - \sigma_z}{2}\right)^2 + \tau_{xz}^2}\,. \tag{D6}$$

*Code availability.* A Python implementation of the present model is publicly available under https://github.com/2phi/weac and https://pypi.org/project/weac (Rosendahl and Weißgraeber, 2022).

*Author contributions.* PW and PLR contributed equally to the design and implementation of the model, to the analysis of the results, and to the writing of the manuscript.

*Competing interests.* The authors declare that they have no conflict of interest.

*Acknowledgements.* We acknowledge support by the German Research Foundation and the Open Access Publishing Fund of Technische Universität Darmstadt. We are grateful for the support by Florian Rheinschmidt who provided the finite element reference model.





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
