# Peer review of "A closed-form model for layered snow slabs"

_The Cryosphere, 2022_

## Author Response (AR1)

**Response to reviewer comments on 'TC-2022-140'**

**Review RC1 by Anonymous Referee #1**

Dear reviewer, thank you for your comments and suggestions, which help us to improve our paper. Please find our response to each concern and remark of the review below:

*This manuscript with the title of "A closed-form model for layered snow slabs" presents a closed-form analytical model for the purpose of investigating and predicting the physical processes that lead to the formation of dry-snow slab avalanches. The reviewer found this manuscript has significant issues regarding fundamental methodology, the weak layer failure mechanisms, validation and conclusions. Therefore, the reviewer do not recommend publishing this manuscript considering the following points.*

> *1.) Section 2 in the manuscript presents the proposed model using continuous laminated beam or plate theory to establish a kinematic model of snowpack. This model can only analyse continuous deformation instead of fracture phenomena of snowpack especially in the weak layer. It is well known that the weak layer failure is a mixed mode damage and fracture propagation, which should be a fundamental study in assessing snow avalanches. The reviewer would suggest authors reconsidering the methodology in investigating snowpack fracture using fracture mechanics approach together with laminated plate theory.*

We agree with the reviewer, that dry snow-slab avalanche release and especially the weak layer failure is governed by mixed-mode fracture processes. A fundamental problem of fracture mechanics is that the formation of new crack surfaces creates a new continuum mechanics boundary-value problem. When solved with finite-element methods, this either requires solving a new boundary-value problem with new mesh or methods of representing the newly boundaries with additional degrees of freedom such as discontinuous ansatz function (XFEM), new field quantities (phase field), degradation of element stiffness instead of removal (damage mechanics and cohesive element approaches). The present model tackles this fundamental challenge by avoiding the need for discretization and numerical concepts altogether by using established concepts of structural mechanics that yield closed-form analytical expressions that, in return, allow of near-real-time solutions of the underlying boundary-value problems. This way, they enable fracture mechanical analyses by allowing for the real-time solution of many different boundary-value problems represented by different crack states or crack advancement.
We have demonstrated this type of analysis with concepts of fracture mechanics in previous works [Ros2020a, Ros2020b].

We have added the following clarifications to the manuscript:

[revised manuscript text omitted]

*2.) Figure 1 show a schematic model of layered snowpack, in which a weak layer is modeled as an elastic foundation using two springs with stiffness Kn and Kt. But this manuscript did present how normal and shear stiffness are coupled in the weak layer. It was mentioned bending and extension coupled in snowpack body as a layered beam. But the failure is in the weak layer, studying coupled effects in the weak layer is much more important than the snow slab above the weak layer.*

The model considers weak-layer stiffnesses Kn and Kt independently. This allows for the consideration of isotropic weak layers (where Kn and Kt are calculated from Young's modulus and Poisson's ratio such that Kn and Kt are coupled through the latter) and for the consideration of anisotropic weak layers whose normal and shear stiffnesses are measured independently.

Weak-layer normal and shear deformations (not stiffnesses) are coupled in the model through their attachment to the slab. That is, weak-layer shear deformations introduce slab bending, which in turn causes weak-layer normal stresses.

We have added the following clarification to the text:

The slab with total thickness $h$ is composed of $N$ layers with individual ply thicknesses $h_i = z_{i+1} - z_i$, each assumed homogeneous and isotropic (Fig. 2). Young's modulus, Poisson's ratio and density of each layer are denoted by $E_i$, $\nu_i$ and $\rho_i$, respectively. The weak layer of thickness $t$ can be anisotropic and its normal and tangential stiffnesses are

$$k_\mathrm{n} = \frac{E'_\mathrm{wl}}{t}, \tag{1a}$$

where $E'_\mathrm{wl} = E_\mathrm{wl}/(1-\nu^2)$ is the weak layer's plane-strain elastic modulus and

$$k_\mathrm{t} = \frac{G_\mathrm{wl}}{t}, \tag{1b}$$

where $G_\mathrm{wl}$ is the weak layer's plane-strain shear modulus, respectively. To account for anisotropic weak layers, these constants can be defined from independent stiffness properties. It is to note, that since the weak layer is connected to the slab, an intrinsic coupling of shear and normal deformation of the weak layer occurs even when the stiffnesses $k_\mathrm{n}$ and $k_\mathrm{t}$ are defined independently.

*3.) Figure 4 shows some different snowpack cases with initial cracks. Fig. 4b is an upslope PST. This is a major case regarding snowpack propagation saw test (PST), some researchers have done experimental work and modelling simulation. The following published papers are for authors' reference:*

*a) Gaume, T. Gast, J. Teran, A. van Herwijnen & C. Jiang, Dynamic anticrack propagation in snow, Nat. Commun. DOI: 10.1038/s41467-018-05181-w.*
*b) Gaume, A. van Herwijnen, G. Chambon, N. Wever, J. Schweizer, Snow fracture in relation to slab avalanche release: critical state for the onset of crack propagation, The Cryosphere 11 (2017) 217–228, https://doi.org/10.5194/tc-11-217-2017.*
*c) Jiye Chen, Blair Fyffe, Dawei Han and Shangtong Yang, Predicting mixed mode damage propagation in snowpack using the extended cohesive damage element method, Theoretical and Applied Fracture Mechanics 122 (2022) 103567, https://doi.org/10.1016/j.tafmec.2022.103567.*

*Reference c summaries the weak layer's fracture mechanisms in the upslope PST case. It is well accepted to recognise a mixed mode damage and fracture in the weak layer under compression and bending. This can also be seen from Fig. 4b, the left part of segment 2 is under compression and shear and the right part under tension and shear. The left part is*

*subjected to both compression and bending in the same way in compression. The bending caused tension and compression have opposite effects on the right part. The left part of segment 2 would fail by compressive crushing and shearing. Once the weak layer fails by the mixed failure mode with crushing and shearing, the snow slab moves down due to gravity. Reference c proposed a new mixed mode damage criteria and successfully assessed the weak layer's failure initiation and propagation. But this manuscript concluded that the upslope PST is a mode I crack (opening). This is questionable. This manuscript has not yet considered the micro damage phenomenon in the weak layer because of the continuous mechanics model used, which cannot reflect the reality of failure in the weak layer. Snow is a porous material with micro crystal construction, which can be easily crushed by compression.*

When we speak of mode I anticrack propagation we refer to the symmetric nature of the deformation field [Bro1999]. Because we consider anticracking, mode I corresponds to weak-layer crushing and collapse. This agrees with the reviewer's understanding of the fracture process described above. Further details of this definition are given in a previous The Cryosphere publication by the authors [Ros2020b].

We have added the following clarifications to the manuscript:

The  model can be used to determine the energy release rate of cracks. Here, we make use of the concept of anticracks (Fletcher and Pollard, 1981), that allows for studying failure of a weak layer in a snowpack exhibiting collapse (Heierli et al., 2008). As typical for fracture mechanics (Broberg, 1989), the symmetry of the displacement field around the crack tip can be used to identify symmetric (mode I) and antisymmetric deformations (mode II). We follow this convention to study mode I (crack closure) and mode II (crack  sliding) energy release rates of anticracks. Further implications are discussed in Rosendahl and Weißgraeber (2020d). Following Krenk (1992), the energy release rate of cracks in weak interfaces can be given as

$$\mathcal{G}(a) = \mathcal{G}_{\mathrm{I}}(a) + \mathcal{G}_{\mathrm{II}}(a) = \frac{\sigma(a)^2}{2k_{\mathrm{n}}} + \frac{\tau(a)^2}{2k_{\mathrm{t}}}, \qquad (19)$$

where $a$ denotes the crack-tip coordinate.  The limitations of the weak-interface kinematics yield energy release rates that cannot capture very short cracks but, again, provide accurate results for cracks of a  minimum length (Hübsch et al., 2021). Cracks shorter than a few millimeters cannot be studied by the present approach.

*4.) Equation 19 and Figure 16 present a total energy release rate in the weak layer: G = G1 + G2. But this manuscript has not used the G for assessing crack propagation. Theoretically, total G is not total fracture energy criterion Gc in the mixed mode case, it contains contributions from G1c and G2cin a coupled model. This important point was missed by this manuscript.*

Of course we agree that mixed-mode crack nucleation and propagation is governed by the interaction law of the mode I and mode II fracture toughnesses as discussed [Ber2023]. However, we are not

aware of any experimental work by other authors that provides this relation. On the contrary, the magnitude of the mode I and II fracture toughnesses in pure modes are not well defined. We are currently working on new methods to identify mixed-mode fracture envelopes of weak interfaces [Ada2022].

However, it is not the aim of the present work to establish a novel failure criterion. We have proposed a mixed-mode finite fracture mechanics approach for non-layered slabs in an earlier work [Ros2020b] and we will incorporate the present model for layered slabs in this framework in the future.

We hope that with the modification in our manuscript in response to your remark 1) this is more clear now.

> *5.) On page 9, the last sentence: " Energy release rates obtained using weak-interface kinematics cannot capture very short cracks …. ". This is a case from the kinematic model. However, if energy release rate obtained by fracture mechanics in weak layer, it can certainly capture any crack length. This has been done in standard fracture benchmarks of laminated composites in literatures.*

We agree with the reviewer that this comprises a limitation of the model. This is the reason why we indicate this as the limitations within the chosen modeling approach. In the present case, the chosen kinematics do not capture extremely short cracks well. However, many established failure analysis concepts that use stress, strain, strain-energy density evaluation at a distance or the concept of finite fracture mechanics, do not make use of the localized quantities so that this limitation does not necessarily render a problem.

We have added the following clarifications – in particular with reference to nonlocal methods – to the text:

The  model can be used to determine the energy release rate of cracks. Here, we make use of the concept of anticracks (Fletcher and Pollard, 1981), that allows for studying failure of a weak layer in a snowpack exhibiting collapse (Heierli et al., 2008). As typical for fracture mechanics (Broberg, 1989), the symmetry of the displacement field around the crack tip can be used to identify symmetric (mode I) and antisymmetric deformations (mode II). We follow this convention to study mode I (crack closure) and mode II (crack   sliding) energy release rates of anticracks. Further implications are discussed in Rosendahl and Weißgraeber (2020d). Following Krenk (1992), the energy release rate of cracks in weak interfaces can be given as

$$\mathcal{G}(a) = \mathcal{G}_{\mathrm{I}}(a) + \mathcal{G}_{\mathrm{II}}(a) = \frac{\sigma(a)^2}{2k_{\mathrm{n}}} + \frac{\tau(a)^2}{2k_{\mathrm{t}}}, \qquad (19)$$

where $a$ denotes the crack-tip coordinate.  The limitations of the weak-interface kinematics yield energy release rates that cannot capture very short cracks but, again, provide accurate results for cracks of a  minimum length (Hübsch et al., 2021). Cracks shorter than a few millimeters cannot be studied by the present approach.

Kinks in the model solution originate from the loading discontinuity introduced by the concentrated skier force. They are a direct result of the plate-theory modeling approach. The agreement with the FEA reference solution is close for all types of investigated profiles and layering effects on weak-layer stress distributions are well captured. Only for profile ⊐ C, the present solution slightly underestimates the normal stress peak directly below the skier. As  we argue in Rosendahl and Weißgraeber (2020b), this observation is  not relevant for the prediction of weak-layer failure  in a snow cover. To study size effects present in any structure,  a nonlocal evaluation of stresses  must be used (Neuber, 1936; Peterson, 1938; Waddoups et al.. This has been discussed in detail by Leguillon (2002), laying the foundation for the successful application of finite fracture mechanics approaches with weak-interface models (Weißgraeber et al., 2015; Rosendahl et al., 2019). Effects of bending stiffness (Fig. 8c vs. d) or bending–extension coupling (Fig. 8e vs. f) resulting from different layering orders, will be discussed in detail below.

The model does not account for contact of the slab with base layers or the remains of a collapsed weak layer. For long weak-layer cracks, the corresponding normal deformations may become too large to be rendered correctly in the present model. A corresponding extension of the present model is work in progress and will allow for the analysis of sustained anticrack growth. As discussed, the weak interface concept used brings the limitation that cracks shorter than a few millimeters cannot be studied.

*6.) Section 3 in this manuscript presents validation. It compared the outcomes from the proposed analytical model with FEM based continuous deformation analysis regarding deformation and stresses. There is no usage of test results in validation. This is also questionable. Firstly, the modelling results from FEM based continuous analysis are not reliable when the FEM mesh with cracks, or without validation by test work, the FEM results would not be accepted to validate other new models regarding fracture issues.*

The use of FEM results to compare two solution methods of the same boundary value problem of structural mechanics models is common practice [Zie2005]. When composed carefully, FE models are very well suited for boundary-value problems including cracks, notches, or other stress concentrators [Kun2013].
The FEM model used in the present work has been conducted carefully, including a mesh sensitivity analysis to rule out mesh effects. Since the focus of the present work is the development of a real-time closed-form model for layered snowpacks and since we use FEM just for validation purposes, we do not believe the work would benefit from a detailed description of the FE model. Instead, we will add a reference to a previous publication, that discusses the FE model in detail – with the only difference being the slab layering.

Moreover, we validate the displacement field produced by our model using full-field DIC displacement measurements in Fig 7. (anonymous reviewer #2 agrees).

We have added the following reference with details on the FE model to the text:

To validate the model, in particular with respect to different slab layerings, we compare the analytical solution to finite element analyses (FEA). The finite element model is assembled from individual layers with unit out-of-plane width on an inclined slope. Each layer is discretized using at least 10 eight-node biquadratic plane-strain continuum elements with reduced integration through its thickness. The lowest layer corresponds to the weak layer and rests on a rigid foundation. Weak-layer cracks are introduced by removing all weak-layer elements on the crack length $a$. The mesh is refined towards stress concentration such as crack tips and mesh convergence has been controlled carefully (Rosendahl and Weißgraeber, 2020c). The weight of the snowpack is introduced by providing the gravitational acceleration $g$ and assigning each layer its corresponding density $\rho$. The load introduced by a skier is modeled as a concentrated force acting on the top of the slab. If skier loading is considered, the horizontal dimensions of the model are chosen large enough for all gradients to vanish. Typically $10\,\mathrm{m}$ suffice. Boundary conditions of PST experiments are free ends. In the FE model, the energy release rate of weak-layer cracks

$$\mathcal{G}_{\mathrm{FE}}(a) = -\frac{\partial \Pi(a)}{\partial a} \approx -\frac{\Pi(a + \Delta a) - \Pi(a - \Delta a)}{2\Delta a}, \quad (21)$$

is computed using the central difference quotient to approximate the first derivative of the total potential $\Pi$ with respect to $a$. The crack increment $\Delta a$ corresponds to the element size and could be increased twofold or threefold without impacting computed values of $\mathcal{G}_{\mathrm{FE}}(a)$. Weak-layer stresses are evaluated in its vertical center.

*7.) On page 12, the last paragraph, " Experimental validations are challenging since direct measurements of stresses are not possible and displacement measurements require considerable experimental effort ". This is truth. However, above references a and b reported critical initial crack length in PST experimental work, which causes crack propagation and sliding of snow slab above weak layer, and the critical initial crack length related bending moment can be used for validation if the critical load can be provided by the analytical model in this manuscript. But this manuscript failed at the validation in this way. Abovementioned reference c has completed a similar validation.*

Thank you for this suggestion. We have incorporated an estimate of the weak-layer (crushing) fracture toughness that can be identified from PST data with the following figure and corresponding text:

Figure 10 shows weak-layer fracture toughnesses determined from critical cut lengths of PSTs with layered slabs throughout the 2019 winter season using the present model. Details of the tests are reported by Bergfeld et al. (2023a,b). The authors performed 21 tests on the same weak layer. While we observe small weak-layer fracture toughnesses at the beginning of January 2019, it quickly increases with the most significant precipitation event in mid January and then remains comparatively constant throughout the rest of the season. For details on the temporal evolution of slab and weak-layer properties, the interested reader is referred to Bergfeld et al. (2023b). For the purpose of validation of the present model, it is to note that all fracture toughnesses computed from the experiments lie within the bounds of the to date lowest and highest published values, $0.01\,\mathrm{J/m^2}$ (Gauthier and Jamieson, 2010) and $2.7\,\mathrm{J/m^2}$ (van Herwijnen et al., 2016), respectively.

The present model can be classified as a structural mechanics model as frequently employed in fracture mechanics. As shown by Bergfeld et al. (2021b), structural models can be used to obtain effective quantities characterizing weak layers. Effective quantities of fracture mechanics models always include microscopic mechanisms without further resolving their microscopic nature Broberg (1989).

[Figure]

**Figure 10.** Weak-layer fracture toughness determined with the present model from critical cut lengths of 21 flat-field propagation saw tests (PSTs) throughout the 2019 winter season on the same surface-hoar weak layer covered by a layered slab of changing thickness (Bergfeld et al., 2023a,b). All results are within the hatched boundaries indicating the thus far lowest and highest published fracture toughness of weak layers, $0.01\,\mathrm{J/m^2}$ (Gauthier and Jamieson, 2010) and $2.7\,\mathrm{J/m^2}$ (van Herwijnen et al., 2016), respectively.

*8.) In example section, manuscript presents a lot of information about stress and deformation distributions in snow slab in different cases. The reviewer thinks this information is less important in assessing snowpack fracture. The outcomes from this proposed model are not only difficult for validation and also in application for assessing snowpack fracture initiation and propagation. As a closed-form solution, it is expected to provide precise failure response which shows a load-deformation curve. Thus, it can clearly present elastic stage, damage point, damage accumulation and final crack point.*

The stress distributions and energy release rates of snowpacks with intact and collapsed weak layers are fundamental for the analysis of weak-layer failure [Sch2003, Sch2016]. We agree that a closed-form analytical solution providing a precise failure response of a slab-weak-layer configuration is of great interest. This is precisely what our model aims at with great efficiency. We provide it as public source-code to other researchers on github.com.

Whether brittle failure phenomena such as weak-layer fracture really show damage accumulation is worth a long discussion. For instance, does damage accumulate in the weak layer if many skiers travel over the same point? Does that mean we should avoid well-traveled slopes – which is certainly counterintuitive?

> *9.) This manuscript mentioned a basic concept of a failure mode in weak layer: mode I (collapse) or anticrack. Although this terminology was used by some researchers in their early work. The reviewer thinks that this terminology is not in the line of the basic concept of fracture mechanics about mode I: opening crack. The wording of collapse or anticrack would not be a right description of the failure mode in opposite way of mode I. Abovementioned reference c properly suggested a terminology of crushing damage by compression, and proposed a mixed mode damage criteria with crushing damage and shearing crack (mode II) to assess the failure in weak layer under mixed loading.*

As discussed in point 3.), we have used the usual way to consider symmetry of the displacement field to distinguish different loading modes of a crack [Bro1999]. This has been done by many researchers that studied anticracks, e.g. in snow [Hei2008, Gau2018], in compaction bands of limestone [Ste2005], pressure (dis)solution in rocks [Fle1981]. We follow the highly influential thoughts of Heierli et al. [Hei2008] in our denomination "collapse" but agree that crushing is an accurate description of the phenomenon, too.

We hope that our additions to the manuscript in response to remark 3.) make this clear in the text.

> *10.) A closed-from should clearly present the failed load and residual strength when crack propagates through the failure response or load-deformation curve. The reviewer thinks that this manuscript has not reach the purpose of investigating and predicting the physical processes that lead to the formation of dry-snow slab avalanches.*

We agree with the reviewer that some of these features would be of interest in a failure model. The present model is not a failure model.

**References used in the response**

[Ada2022] Adam V., Bergfeld B., van Herwijnen A., Weißgraeber P., Rosendahl P. L. (2022) Modified propagation saw tests for analyses of weak-layer fracture properties (https://www.igsoc.org/wp-content/uploads/2022/10/procabstracts_79.html#A3837)

[Baz2003] Bažant, Z. P., Zi, G., & McClung, D. (2003). Size effect law and fracture mechanics of the triggering of dry snow slab avalanches. Journal of Geophysical Research: Solid Earth, 108(B2).

[Ber2023] Bergfeld, B., van Herwijnen, A., Bobillier, G., Rosendahl, P. L., Weißgraeber, P., Adam, V., ... & Schweizer, J. (2023). Temporal evolution of crack propagation characteristics in a weak snowpack layer: conditions of crack arrest and sustained propagation. *Natural Hazards and Earth System Sciences*, *23*(1), 293-315.

[Bro1999] Broberg, K. B. (1999). Cracks and fracture. Elsevier.

[Fle1981] Fletcher, R. C., and D. D. Pollard (1981), Anticrack model for pressure solution surfaces, Geology, 9, 419–424.

[Gau2018] Gaume, J., Gast, T., Teran, J., Van Herwijnen, A., & Jiang, C. (2018). Dynamic anticrack propagation in snow. Nature communications, 9(1), 3047.

[Hei2008] Heierli, J., Gumbsch, P., & Zaiser, M. (2008). Anticrack nucleation as triggering mechanism for snow slab avalanches. *Science*, *321*(5886), 240-243.

[Kun2013] Kuna, M. (2013). Finite elements in fracture mechanics. *Solid Mechanics and Its Applications*, *201*, 153-192.

[Ros2020a] Rosendahl, P. L., & Weißgraeber, P. (2020). Modeling snow slab avalanches caused by weak-layer failure–Part 1: Slabs on compliant and collapsible weak layers. *The Cryosphere*, *14*(1), 115-130.

[Ros2020b] Rosendahl, P. L., & Weißgraeber, P. (2020). Modeling snow slab avalanches caused by weak-layer failure–Part 2: Coupled mixed-mode criterion for skier-triggered anticracks. *The Cryosphere*, *14*(1), 131-145.

[Sch2003] Schweizer, J., Bruce Jamieson, J., & Schneebeli, M. (2003). Snow avalanche formation. *Reviews of Geophysics*, *41*(4).

[Sch2011] Schweizer, J., van Herwijnen, A., & Reuter, B. (2011). Measurements of weak layer fracture energy. Cold Regions Science and Technology, 69(2-3), 139-144.

[Sch2016] Schweizer, J., Reuter, B., Van Herwijnen, A., & Gaume, J. (2016, October). Avalanche release 101. In Proceedings ISSW (pp. 1-11).

[Ste2005] Sternlof, K. R., Rudnicki, J. W., & Pollard, D. D. (2005). Anticrack inclusion model for compaction bands in sandstone. Journal of Geophysical Research: Solid Earth, 110(B11).

[Zie2005] Zienkiewicz, O. C., & Taylor, R. L. (2005). *The finite element method for solid and structural mechanics*. Elsevier.

**CC1 by Xingyue Li**

Dear reviewer, thank you for your comments and suggestions, which help us to improve our paper. Please find our response to each concern and remark of the review below:

*This study develops an analytical model for the investigation of mechanical behavior of a stratified snow cover over a weak layer. The snow cover is considered as an arbitrarily layered beam solved by laminate mechanics, while the weak layer is modeled as a set of springs attached to the bottom of the snow cover. The model is firstly verified with the experimental data and the numerical results from finite element modeling, and then adopted to investigate the factors affecting slab release, including layering, bridging and slope angle. The key novelty of this study to the reviewer is the consideration of the layered snow, compared to the previous study conducted by the authors. The outcomes from this study could offer useful information for understanding the failure behavior of stratified snow over a weak layer, and thus give relevant information on slab avalanche release. However, there are some concerns to be clarified as detailed below.*

> *1) Consideration of the crack: As described in the methodology, the model considers the weak layer as springs with normal and shear stiffnesses, and can handle different scenarios such as the ones with partially collapsed weak layer in Fig. 4. But it is not clear how the crack in the weak layer is considered and whether the scenarios in Fig. 4 are predefined. If the scenarios are predefined, how to determine these different initial conditions in practice? If not, please clarify the triggering of the crack. For example, what is the criteria to trigger the crack? Do the springs have certain shear and normal strength, above which they break? Please also discuss the propagation of the crack with time.*

The model can be used to study both slabs on intact weak layers and slabs where the weak layer has partially failed. The weak layers are modeled with a so-called weak-interface approach as it has been used in many works to study structural situation with stiffer structures being supported by foundations of significantly lower stiffness [Het1946]. Such weak-interface models can also be used to study locally failed foundations by removing the support effect of the interface [Kre1992, Len2001]. The present model can then be used to establish failure models that require the stress distribution in the weak interface and the energy release rate of crack configurations [Ste2015]. We have proposed a mixed-mode finite fracture mechanics approach for non-layered slabs in an earlier work [Ros2020b] and we will incorporate the present model for layered slabs in this framework in the future.

We thank the reviewer for pointing out that the description of the boundary conditions in both scenarios (slabs on intact weak layers and slabs where the weak layer has partially failed) should be improved.

We have revised the section to make this more clear:

 To study situations where the weak layer has partially failed, the case of an unsupported slab must be considered. The situation can occur when the weak layer has collapsed or when a saw cut is introduced in a propagation saw test. Accounting for such cases allows for the use of the present model in failure models for anticrack nucleation (Rosendahl and Weißgraeber, 2020d) or growth (Bergfeld et al., 2021b). If the slab is not supported by an elastic foundation, the general solution simplifies. In the equilibrium conditions (6a) to (6c), the normal and shear stress terms are omitted since no stresses act on the bottom side of the slab. The constitutive equations (7a) and (7b) remain the same. After some calculation (see Appendix B) one obtains the general solution of polynomials of fourth order. In matrix form, the system reads

$$\boldsymbol{z}_\circ(x) = \boldsymbol{\mathcal{P}}(x)\, \boldsymbol{c}_\circ + \boldsymbol{p}(x), \tag{16}$$

where $\boldsymbol{\mathcal{P}}(x)$ and $\boldsymbol{p}(x)$ are the polynomial matrix and vector, respectively. Again, a vector of six unknown coefficients

$$\boldsymbol{c}_\circ = \begin{bmatrix} C_\circ^{(1)} & C_\circ^{(2)} & \dots & C_\circ^{(6)} \end{bmatrix}^\mathsf{T}. \tag{17}$$

must be determined from boundary and transmission conditions.

*2) Comparison with the results from a homogeneous equivalent layer: In the results 4.1, the current model has been compared with the homogeneous model by Monti et al. (2015). It is stated that "Both concepts are benchmarked against the stiffnesses computed using finite element analyses", please clarify whether "both concepts" are the current model and the model by Monti et al. (2015). If yes, as they have been benchmarked already, why the model by Monti et al. (2015) does not have consistent stiffness with the FEA in Fig. 10? In addition, please clarify that if the correct equivalent stiffnesses are implemented to a homogeneous model such as that by Monti et al. (2015) or the previous model by the authors (Rosendahl and Weißgraeber, 2020a), will the homogeneous models give good prediction on the mechanical behavior of the slab?*

Thank you for pointing out this ambiguity. What we mean is that we compare the model by Monti et al. against finite element analyses and also compare the present model against FE analyses. We have clarified this in the text in the following way:

Table 3 and Fig. 11 compare stiffnesses computed with the present concept of laminate mechanics, Eqs. (8a) to (8d), with these stiffnesses of an equivalent homogeneous slab computed with properties obtained from the equivalence concept, Eqs. (23a) to (23d).  Table 3 and Fig. 11 compare both concepts against the stiffnesses computed using finite element analyses. Here, the corresponding stiffnesses are obtained from the force response of unit extension and bending deformations. While Eqs. (8a) to (8d) reproduce the reference stiffnesses exactly, the equivalent layer approach systematically underestimates the extensional, the bending, and the shear stiffnesses and cannot account for bending–extension couplings.

Monti et al. have not focused on stiffnesses but instead looked at stresses within the snowpack, where the agreement was not 100% but deemed satisfying by the authors. We can only speculate that they were not aware of the stiffness discrepancies.

If slabs are layered, there is no one equivalent stiffness. Instead, layered slabs always respond differently to bending or tension. When there are only small differences in the stiffness of all layers of the slab, the use of an equivalent stiffness can provide satisfactory results because the error from neglecting layering effects is small [Ber2023].

We aim at clarifying Monti's intentions with the following addition to the manuscript:

> The mechanical behavior of the slab is governed by its stiffnesses. A layered system may have different stiffnesses with respect to extension, shear, or bending. Hence, we distinguish the extensional stiffness $A_{11}$, the bending–extension coupling stiffness $B_{11}$, the bending stiffness $D_{11}$, and the shear stiffness $A_{55}$. They are obtained from integration of the individual layer stiffnesses as specified in Eqs. (8a) to (8d). The ordering of layers influences each stiffness differently. That is, the simple homogenization of layered continua in
>
> the form of a single homogeneous equivalent layer is insufficient.  With the aim to describe the shear stresses in a slab, Monti et al. (2015) proposed a concept of equivalent layers to allow for the use of Boussinesq's solution for an isotropic elastic half-plane. They followed concepts developed in order to describe the surface deformation of layered systems in normal direction (De Barros, 1966). Using the equivalent Young's modulus $E_{eq}$ introduced by Monti et al. (2015), the stiffnesses of a homoge-

**References used in the response**

[Ber2023] Bergfeld, B., van Herwijnen, A., Bobillier, G., Rosendahl, P. L., Weißgraeber, P., Adam, V., ... & Schweizer, J. (2023). Temporal evolution of crack propagation characteristics in a weak snowpack layer: conditions of crack arrest and sustained propagation. *Natural Hazards and Earth System Sciences*, *23*(1), 293-315.

[Het1946] Hetényi, M., & Hetbenyi, M. I. (1946). *Beams on elastic foundation: theory with applications in the fields of civil and mechanical engineering* (Vol. 16). Ann Arbor, MI: University of Michigan press.

[Kre1992] Krenk, S. (1992). Energy release rate of symmetric adhesive joints. *Engineering Fracture Mechanics*, *43*(4), 549-559.

[Len2001] Lenci, S. (2001). Analysis of a crack at a weak interface. *International Journal of Fracture*, *108*(3), 275-290.

[Ste2015] Stein, N., Weißgraeber, P., & Becker, W. (2015). A model for brittle failure in adhesive lap joints of arbitrary joint configuration. Composite Structures, 133, 707-718.

[Ros2020b] Rosendahl, P. L., & Weißgraeber, P. (2020). Modeling snow slab avalanches caused by weak-layer failure–Part 2: Coupled mixed-mode criterion for skier-triggered anticracks. *The Cryosphere*, *14*(1), 131-145.

**RC2 by Anonymous Referee #2**

The review is identical to CC1 by by Xingyue Li.

**RC3 by Anonymous Referee #3**

Dear reviewer, thank you for your comments and suggestions, which help us to improve our paper. Please find our response to each concern and remark of the review below:

*The Authors propose a closed-form analytical model for the mechanical behavior of stratified snow covers adopting the first-order shear deformation theory of laminated plates under cylindrical bending. The weak layer is modelled as an infinite set of smeared springs with normal and shear stiffness at the base of the plate. The problem is worth to be studied and it is of large interest for the scientific community. There are some points that must be clarified and further investigated.*

> *1. It is not clear the remind to De Saint Venant's principle (l. 246) and how it can be applied in the model proposed by the authors as the localized stresses and effects are of primary importance for the release of the avalanche.*

The question is well answered by Wikipedia: "Saint-Venant's principle, named after Adhémar Jean Claude Barré de Saint-Venant, a French elasticity theorist, may be expressed as follows: the difference between the effects of two different but statically equivalent loads becomes very small at sufficiently large distances from load." (https://en.wikipedia.org/wiki/Saint-Venant%27s_principle).

For the model this means that introducing the skier loading as distributed loads (width of the skis) or concentrated into one force (statically equivalent) has the same effect on weak-layer stresses.

We have included the following clarification in the revised manuscript:

> Individual segments are connected through transmission conditions given in terms of displacements and section forces (see Appendix C). Adding boundary conditions at the left and right ends of the beam, assembles the desired global system. Since localized loads (e.g., skier weight) are introduced as a (statically equivalent) change of the section forces, the solution will not be able to fully render effects in the close vicinity of the load introduction. This is discussed in the validation in Section 3.2.

> *2. The authors must clarify how the concentrated load applied on the surface of the slab enters into the solution of the first-order system of Eqn. (11).*

The ODE system is obtained by using the constitutive equations and condition of equilibrium of the slab-weak-layer system (Appendix A and B in the manuscript). The concentrated loads are introduced in the next step, when boundary and transmission conditions are considered (see Appendix C for details). A concentrated force introduces a discontinuity in the equilibrium of lateral forces and, hence, requires a transmission condition.

We have added the following clarifications:

External concentrated forces (e.g., skiers) are introduced  with their normal and tangential components $F_n$ and $F_t$ and with their resulting moment $M = -hF_t/2$.  They have to be accounted for in the form of the transmission conditions between two segments

$$\Delta N = F_t, \quad \Delta V = F_n, \quad \Delta M = -\frac{h}{2}F_t, \qquad (C4)$$

where again, the $\Delta$ operator expresses the difference between left and right segments. Therefore, at points of such loads the slab must always be split into segments to allow for the definition of the transmission conditions.

*3. The authors must clarify how the size of the crack tip can be effectively measured in a real snowpack as it affects the results of the stability analysis.*

In the case of the most relevant experiment for analysis of the fracture behavior of weak layers – the propagation saw test – the crack length corresponds to the critical cut length at which the PST fails.

We added references to Bergfeld et al. (2023a,b) for details on how PSTs are performed and evaluated:

Figure 10 shows weak-layer fracture toughnesses determined from critical cut lengths of PSTs with layered

slabs throughout the 2019 winter season using the present model. Details of the tests are reported by Bergfeld et al. (2023a,b). The authors performed 21 tests on the same weak layer. While we observe small weak-layer fracture toughnesses at the beginning of January 2019, it quickly increases with the most significant precipitation event in mid January and then remains comparatively constant throughout the rest of the season. For details on the temporal evolution of slab and weak-layer properties, the interested reader can refer to Bergfeld et al. (2023b). For the purpose of validation of the present model, it is to note that all fracture toughnesses computed from the experiments lie within the bounds of the to date lowest and highest published values, $0.01\,\mathrm{J/m^2}$ (Gauthier and Jamieson, 2010) and $2.7\,\mathrm{J/m^2}$ (van Herwijnen et al., 2016), respectively.

*4. The authors confuse the energy release rate with the failure criterion. To let the crack to propagate, it is necessary that the G term equals the critical G in the mixed mode. It is worth to mention that, in fracture mechanics, Mode 1 refers to an opening mode, differently from what the authors state in their approach. This point must be addressed in order to avoid misunderstandings.*

The present work does not consider failure or a failure criterion but aims at providing all quantities necessary for a failure assessment. In our view, the most important ingredients are weak-layer stresses (normal and shear) and weak-layer energy release rates (collapse and sliding). Eq. (19) provides the equations for the latter. That is, Eq. (19) enables the formulation of mixed-mode weak-layer failure criteria.

An example of how mode I and mode II energy release rates can be considered individually is given in Figure 16 that analyzes the effect of slope angle on the mode mixity of the energy release rates in propagation saw tests.

Note that we have proposed a mixed-mode finite fracture mechanics approach for non-layered slabs in an earlier work [Ros2020b] and we will incorporate the present model for layered slabs in this framework in the future.

When we speak of mode I anticrack propagation, we refer to the symmetric nature of the deformation field [Bro1999]. Because we consider anticracking, mode I corresponds to weak-layer crushing and collapse. Further details of this definition are given in a previous The Cryosphere publication by the authors [Ros2020b].

We have added the following clarifications to make this more clear.

The  model can be used to determine the energy release rate of cracks. Here, we make use of the concept of anticracks (Fletcher and Pollard, 1981), that allows for studying failure of a weak layer in a snowpack exhibiting collapse (Heierli et al., 2008). As typical for fracture mechanics (Broberg, 1989), the symmetry of the displacement field around the crack tip can be used to identify symmetric (mode I) and antisymmetric deformations (mode II). We follow this convention to study mode I (crack closure) and mode II (crack  sliding) energy release rates of anticracks. Further implications are discussed in Rosendahl and Weißgraeber (2020d). Following Krenk (1992), the energy release rate of cracks in weak interfaces can be given as

$$\mathcal{G}(a) = \mathcal{G}_{\mathrm{I}}(a) + \mathcal{G}_{\mathrm{II}}(a) = \frac{\sigma(a)^2}{2k_{\mathrm{n}}} + \frac{\tau(a)^2}{2k_{\mathrm{t}}}, \qquad (19)$$

where $a$ denotes the crack-tip coordinate.  The limitations of the weak-interface kinematics yield energy release rates that cannot capture very short cracks but, again, provide accurate results for cracks of a  minimum length (Hübsch et al., 2021). Cracks shorter than a few millimeters cannot be studied by the present approach.

[…]

Kinks in the model solution originate from the loading discontinuity introduced by the concentrated skier force. They are a direct result of the plate-theory modeling approach. The agreement with the FEA reference solution is close for all types of investigated profiles and layering effects on weak-layer stress distributions are well captured. Only for profile ⊐ C, the present solution slightly underestimates the normal stress peak directly below the skier. As  we argue in Rosendahl and Weißgraeber (2020b), this observation is  not relevant for the prediction of weak-layer failure  in a snow cover. To study size effects present in any structure,  a nonlocal evaluation of stresses  must be used (Neuber, 1936; Peterson, 1938; Waddoups et al. . This has been discussed in detail by Leguillon (2002), laying the foundation for the successful application of finite fracture mechanics approaches with weak-interface models (Weißgraeber et al., 2015; Rosendahl et al., 2019). Effects of bending stiffness (Fig. 8c vs. d) or bending–extension coupling (Fig. 8e vs. f) resulting from different layering orders, will be discussed in detail below.

[…]

The model does not account for contact of the slab with base layers or the remains of a collapsed weak layer. For long weak-layer cracks, the corresponding normal deformations may become too large to be rendered correctly in the present model. A corresponding extension of the present model is work in progress and will allow for the analysis of sustained anticrack growth. As discussed, the weak interface concept used brings the limitation that cracks shorter than a few millimeters cannot be studied.

**References used in the response:**

[Bro1999] Broberg, K. B. (1999). Cracks and fracture. Elsevier.

[Ros2020b] Rosendahl, P. L., & Weißgraeber, P. (2020). Modeling snow slab avalanches caused by weak-layer failure–Part 2: Coupled mixed-mode criterion for skier-triggered anticracks. *The Cryosphere*, *14*(1), 131-145.

---

## Author Response (AR2)

**Response to reviewer comments on the revised manuscript of 'TC-2022-140'**

**Report #1 by Anonymous Referee #3**

Dear reviewer, thank you for reviewing our revised manuscript. Please find our response to your remark below:

*The authors addressed the observations raised by the reviewer. Anyway, to reviewers opinion, the authors must further stress that their work does not consider failure or a failure criterion but aims at providing all quantities necessary for a failure assessment.*

We have added the following paragraph at the end of section 1 defining the scope of the present work before presenting the model development in section 2. We hope this now conclusively explains the aims and limitations of the manuscript.

> In order to account for the crucial effect of layering on failure processes within a snowpack, we propose a new model for layered snow slabs on collapsible weak layers. Using the concepts of mechanics of layered composites (Jones, 1998) and weak interfaces (Lenci, 2001), we provide closed-form expressions that allow for real-time computations of snowpack deformations, weak-layer stresses, and the energy release rate of cracks in the weak layer. The work aims at establishing a fast computational framework for the physical analysis of the fracture process that leads to the formation of snow slab avalanches. For this purpose, the model considers discrete configurations of layered slabs supported by a weak layer that have collapsed on a given length. We to not attempt to formulate weak-layer failure criteria or to simulate crack advance but aim at providing the mathematical tools for such exercises.